# A multi-grained symmetric differential equation model for learning protein-ligand binding dynamics

Shengchao Liu [1,7] ✉, Weitao Du[2,7], Hannan Xu [3], Yanjing Li[4], Zhuoxinran Li[5], Vignesh Bhethanabotla [4], Divin Yan[4], Christian Borgs [1,8] ✉, Anima Anandkumar [4,8] ✉, Hongyu Guo [6,8] ✉ & Jennifer Chayes[1,8] ✉

Molecular dynamics (MD) simulation is a key tool in drug discovery for predicting protein-ligand binding affinities, transport properties, and pocket dynamics. While advances in numerical and machine learning (ML) methods have improved MD efficiency, accurately modeling long-timescale dynamics remains challenging. We introduce NeuralMD, an ML surrogate that accelerates and enhances MD simulations of protein-ligand binding. NeuralMD employs a physics-informed, multi-grained, group-symmetric framework comprising (1) BindingNet, which enforces symmetry via vector frames and captures multi-level protein-ligand interactions, and (2) an augmented neural differential equation solver that learns trajectories under Newtonian mechanics. Across ten single-trajectory and three multi-trajectory tasks, NeuralMD achieves up to 15 × lower reconstruction error and 70% higher validity than existing ML baselines. The predicted oscillations closely align with ground-truth dynamics, establishing NeuralMD as a foundation for next-generation protein-ligand simulation research.

The molecular dynamics (MD) simulation for protein-ligand binding is one of the fundamental tasks in drug discovery[1–4]. Such simulations of binding systems are a key component of the drug discovery pipeline to select, refine, and tailor the chemical structures of potential drugs to enhance their efficacy and specificity. To simulate the protein-ligand dynamics, *numerical MD* methods have been extensively developed[5,6]. However, the numerical MD methods are computationally expensive due to the expensive force calculations on individual atoms in a large protein-ligand system.

To alleviate this issue, machine learning (ML) surrogates have been proposed to either augment or replace numerical MD methods to estimate the MD trajectories. However, all prior ML approaches for MD simulation are limited to single-system and not protein-ligand complex[7–10]. A primary reason is the lack of large-scale datasets for protein-ligand binding (the first large-scale dataset with binding dynamics was released in May 2023[11]). Further, prior ML-based MD approaches limit to fitting the energy surface so as to study the MD dynamics on a small timestep (e.g., 1e-15 s)[12–19], while simulation on a larger timestep (e.g., 1e-9 s) is needed for specific tasks, such as detecting the transient and cryptic states in binding dynamics[20]. However, current longer-time ML MD simulations are challenging due to the catastrophic buildup of errors over longer rollouts[21].

Another critical aspect that needs to be considered in ML-based modeling is the group symmetry present in the protein-ligand geometry. Specifically, the geometric function over molecular systems should be equivariant to rotation and translation, i.e., SE(3)-equivariance. One principled approach to satisfy equivariance is to use

[1]University of California Berkeley, Berkeley, CA, US. [2]DAMO Academy, Bellevue, WA, US. [3]University of Oxford, Oxford, UK. [4]California Institute of Technology, Pasadena, CA, US. [5]University of Toronto, Toronto, ON, Canada. [6]National Research Council Canada, Ottawa, ON, Canada. [7]These authors contributed equally: Shengchao Liu, Weitao Du. [8]These authors jointly supervised this work: Christian Borgs, Anima Anandkumar, Hongyu Guo and Jennifer Chayes. ✉e-mail: shengchao.liu@berkeley.edu; borgs@berkeley.edu; anima@caltech.edu; hongyu.guo@uottawa.ca; jchayes@berkeley.edu

vector frames, which have been previously explored for single molecules[22], but not yet for the binding complexes. The vector frame basis achieves SE(3)-equivariance by projecting vectors (e.g., positions and accelerations) to the vector frame basis, and such a projection can maintain the equivariant property with efficient calculations[17].

To this end, we propose NeuralMD, a multi-grained physics-informed approach designed to handle extended-timescale MD simulations in protein-ligand binding. Our multi-grained method explicitly decomposes the large molecular complexes into three granularities to obtain an efficient approach for modeling the large molecular system: the atoms in ligands, the backbone structures in proteins, and the residue-atom pairs in binding complexes. We achieve group symmetry in BindingNet through the incorporation of vector frames, and include three levels of vector frame bases for multi-grained modeling, from the atom and backbone level to the residue level for binding interactions.

Additionally, our ML approach, NeuralMD, leverages data-driven techniques to learn Newtonian mechanics. In MD, the movement of atoms is determined by Newton's second law, $F = m \cdot a$, where $F$ is the force, $m$ is the mass, and $a$ is the acceleration of each atom. By integrating acceleration and velocity with respect to time, we can obtain the velocities and positions, respectively, as shown in Fig. 1. Thus, in NeuralMD, we formulate the trajectory simulation as a second-order ordinary differential equation (ODE) or second-order stochastic differential equation (SDE) problem. We augment derivative space by concurrently calculating the accelerations and velocities, allowing simultaneous integration of velocities and positions.

To evaluate the effectiveness and efficiency of NeuralMD, we design ten single-trajectory and three multi-trajectory binding simulation tasks. For quantitative assessment, we employed two reconstruction metrics and two validity metrics. The results demonstrate that NeuralMD consistently outperforms other ML methods, achieving up to 15 × reduction in reconstruction error and 70% increase in validity. Additionally, for qualitative analysis, we introduced a fluctuation measure to characterize ligand oscillation, supported by the Supplementary Video 1. These results highlight that while NeuralMD is not perfect, it generalizes more effectively than other ML baselines. Finally, in terms of efficiency, NeuralMD delivers up to over 1K × speedup compared to numerical methods.

## Results

### Preliminaries

**Ligand data structure.** In this work, we consider binding complexes involving small molecules as ligands. Small molecules can be treated as sets of atoms in the 3D Euclidean space, $\{f^{(l)}, x^{(l)}\}$, where $f^{(l)}$ and $x^{(l)}$ represent the atomic numbers and 3D Euclidean coordinates for atoms in each ligand, respectively.

**Protein data structure.** Proteins are macromolecules, which are essentially chains of amino acids or residues. There are 20 natural amino acids, and each amino acid is a small molecule. Noticeably, amino acids are made up of three components: a basic amino group (-NH₂), an acidic carboxyl group (-COOH), and an organic R group (or side chain) that is distinct to each amino acid. Additionally, the carbon that connects all three groups is called $C_\alpha$. Due to the large number of atoms in proteins, this work proposes a multi-grained method for modeling protein-ligand complexes. In this regard, the *backbone-level* data structure for each protein is $\{f^{(p)}, \{x_N^{(p)}, x_{C_\alpha}^{(p)}, x_C^{(p)}\}\}$, for the residue type and the coordinates of $N - C_\alpha - C$ in each residue, respectively. We may omit the superscript in the coordinates of backbone atoms, as these backbone structures are different from protein residues. In addition to the backbone level, as a coarser-grained view, we further consider *residue-level* information for modeling binding interactions, $\{f^{(p)}, x^{(p)}\}$, where the coordinate of $C_\alpha$ is taken as the residue-level coordinate, i.e., $x^{(p)} \triangleq x_{C_\alpha}^{(p)}$.

**MD simulations.** Generally, MD describes how each atom in a molecular system moves over time, following Newton's second law of motion:

$$F = m \cdot a = m \cdot \frac{d^2 x}{dt^2}, \tag{1}$$

where $F$ is the force, $m$ is the mass, $a$ is the acceleration, $x$ is the position, and $t$ is the time. Then, an MD simulation will take a second-order integration to get the trajectories for molecular systems like a small molecule, a protein, a polymer, or a protein-ligand complex. The numerical MD methods can be classified into classical MD and ab-initio MD, where the difference lies in how the force on each atom is calculated: classical MD uses force field approaches to predict the atomic forces[5], while ab-initio MD calculates the forces using quantum

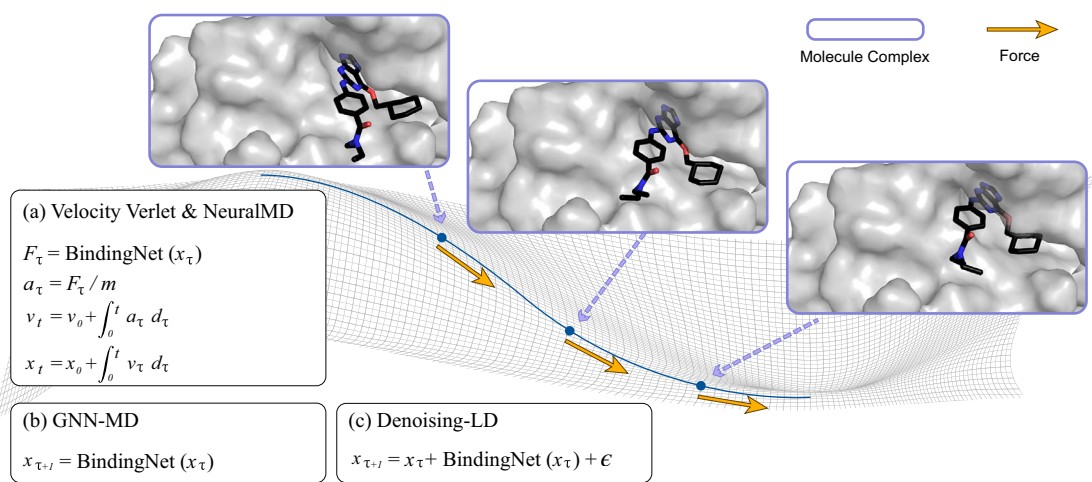

**Fig. 1 | Comparison of simulation approaches for binding dynamics.** Each method aims to evolve a system from a higher-energy state to a stable, low-energy bound complex, visualized as descending a free-energy landscape. **a** Pipeline for Velocity Verlet and NeuralMD (ours). They are the second-order integrals of Newton's equations. Forces are computed at each step to update positions and velocities along a trajectory. **b** Pipeline for GNN-MD. A graph neural network acts as a surrogate model for trajectory prediction. **c** Pipeline for Denoising-LD. It is a diffusion-based approach that iteratively denoises a noisy configuration to directly sample equilibrium structures.

mechanical methods, such as density functional theory (DFT)[6]. More recently, ML MD methods have opened another perspective by utilizing the group symmetric tools for geometric representation and energy prediction[12–19], as well as the automatic differential tools for trajectory learning[23–28]. Please check Supplementary A and Supplementary Table S1 for a more detailed discussion.

**Newtonian dynamics and langevin dynamics.** Numerical MD methods can be additionally divided into two categories: using Newtonian dynamics or Langevin dynamics. Newtonian dynamics is suitable for idealized systems with negligible thermal effects or when deterministic trajectories are required, while Langevin dynamics is adopted where thermal effects play a significant role and when the system is being studied at a finite temperature. In the ML-based MD simulations, adopting Newtonian dynamics or Langevin dynamics can be treated as an option for introducing different inductive biases. In this work, we propose two versions: an ODE solver and a SDE solver concerning Newtonian dynamics and Langevin dynamics, respectively. Noticeably, our experiential dataset, MISATO[11], uses Newtonian dynamics with Langevin thermostats, and the information on solvent molecules is not provided.

**Problem setting: MD simulation in protein-ligand binding.** In this work, we are interested in learning the MD simulation in the protein-ligand binding system, and we consider the semi-flexible setting[29], i.e., proteins with rigid structures and ligands with flexible movements. Thus, the problem is formulated as follows: suppose we have a fixed protein structure $\{f^{(p)}, \{x_N^{(p)}, x_{C_\alpha}^{(p)}, x_C^{(p)}\}\}$ and a ligand with its initial structure and velocity, $\{f^{(l)}, x_0^{(l)}, v_0^{(l)}\}$. We want to predict the trajectories of ligands following the Newtonian dynamics, i.e., the movement of $\{x_t^{(l)}, ...\}$ over time. We also want to clarify two critical points about this problem setting: (1) Our task is trajectory prediction, i.e., positions as labels, and no explicit energy and force are considered as labels. ML methods for energy prediction followed by a numerical ODE/SDE solver require a smaller timestep (e.g., 1e-15 s), while trajectory prediction, which directly predicts the coordinates over time, is agnostic to the magnitude of the timestep. This is appealing for tasks with larger timestep (e.g., 1e-9 s), as will be discussed below. (2) Each trajectory is composed of a series of geometries of molecules, and such geometries are called *snapshots*. We avoid using *frame* since we will introduce the *vector frame* in modeling the binding complex in the following sections.

**Pipeline: neuralMD**
In this section, we briefly introduce NeuralMD, our framework for learning MD simulations in protein-ligand binding. It has two main phases: (1) BindingNet: a multi-grained SE(3)-equivariant geometric model that represents the protein-ligand complex across three granularities. It utilizes vector frame bases at the atom level for ligands, the backbone level for proteins, and the residue level for protein-ligand complexes. (2) Dynamics Solver: either a second-order ODE solver or a second-order SDE, used to model trajectories as Newtonian or Langevin dynamics, respectively. Figure 2 depicts the key components and pipeline of NeuralMD. Please check the Methods Section for more details.

**(1) Multi-grained vector frames.** We consider three data perspectives: atom level for ligands, backbone level for proteins, and residue level for protein-ligand complexes. These perspectives allow us to construct vector frames corresponding to each level, serving as reference bases for SE(3)-geometric modeling.

**(2) Multi-grained SE(3)-equivariant binding modeling: bindingNet.** To model protein-ligand binding prediction, we introduce BindingNet, which leverages the three vector frames established in the first step. The core idea is to achieve rotation-equivariant modeling by projecting vector variables onto the reference frames.

**(3) MD modeling of binding complexes: NeuralMD.** Finally, we leverage the binding energy predictions from BindingNet to simulate protein-ligand binding dynamics. This involves using a second-order ODE for Newtonian dynamics and a SDE for Langevin dynamics. These three steps collectively form the foundation of NeuralMD for modeling binding dynamics.

## Experiment Setting
**Datasets.** One of the main bottlenecks of studying ML for MD simulation in protein-ligand binding is insufficient data. Recently, the community has put more effort into gathering the datasets, and we consider MISATO in our work[11]. It is built on 16,972 experimental protein-ligand complexes extracted from the protein data bank (PDB)[30]. Such data is obtained using X-ray crystallography, Nuclear Magnetic Resonance (NMR), or Cryo-Electron Microscopy (Cryo-EM), where systematic errors are unavoidable. This motivates the MISATO project, which utilizes semi-empirical quantum mechanics for structural curation and refinement, including regularization of the ligand geometry. For each protein-ligand complex, the trajectory comprises 100 snapshots in 8 nanoseconds under fixed temperature and pressure. In Supplementary D (Supplementary Figs. S1 and S2), we list the basic statistics of MISATO, e.g., the number of atoms in small molecule ligands, and the number of residues in proteins.

**Baselines.** Using ML for energy and force prediction, followed by trajectory prediction using numerical integration method, has been widely explored in the community, e.g., HDNNPs[7], DeePMD[9], TorchMD[31], and Allegro-LAMMPS[32]. Here we extend this paradigm for binding dynamics and propose VerletMD, which utilizes BindingNet for energy prediction on each snapshot and velocity Verlet algorithm to get the trajectory. Additionally, we mainly focus on ML methods for trajectory prediction in this work, *i.e.*, no energy or force is considered as labels. GNN-MD is to apply geometric graph neural networks (GNNs) to predict the trajectories in an auto-regressive manner[11,33]. More concretely, GNN-MD takes the coordinates and other molecular information as inputs at time $t$ and predicts the coordinates at time $t + 1$. DenoisingLD (denoising diffusion for Langevin dynamics)[33–35] is a baseline method that models the trajectory prediction as denoising diffusion task[36], and the inference for binding trajectory essentially becomes the Langevin dynamics. CG-MD learns a dynamic GNN and a score GNN[33], which are essentially the hybrid of GNN-MD and DenoisingLD. Here, to make the comparison more explicit, we compare these two methods (GNN-MD and DenoisingLD) separately. We want to highlight that we keep the same backbone model, BindingNet, for force or position prediction for all the baselines and NeuralMD.

**Experiments Settings.** We consider two experimental settings. The first type of experiment is the single-trajectory prediction, where both the training and test data are snapshots from the same trajectory, and they are divided temporally. The second type of experiment is the multi-trajectory prediction, where each data point is the sequence of all the snapshots from one trajectory, and the training and test data correspond to different sets of trajectories. All the experiments are conducted with three random seeds (0, 42, 123), and details are provided in Supplementary E and Supplementary Table S2. The mean results are reported in the main article, and the standard deviations are reported in Supplementary F and Supplementary Tables S3-S6.

**Evaluation Metrics.** For quantitative evaluation, we use two reconstruction metrics and two validity metrics: (1) Reconstruction. For MD simulation, the evaluation is a critical factor for evaluating trajectory prediction. For both experiment settings, the trajectory reconstruction is the most straightforward evaluation metric. To evaluate this, we

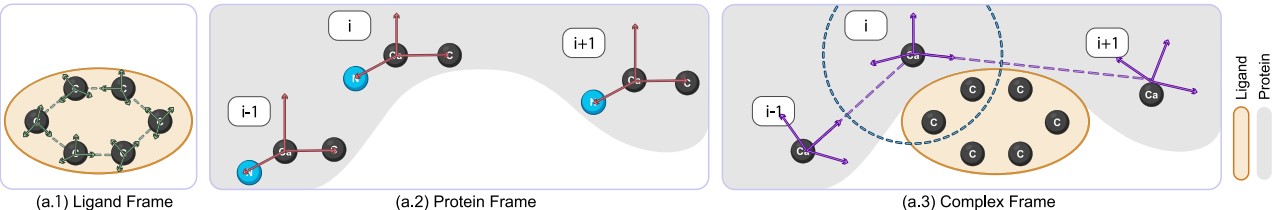

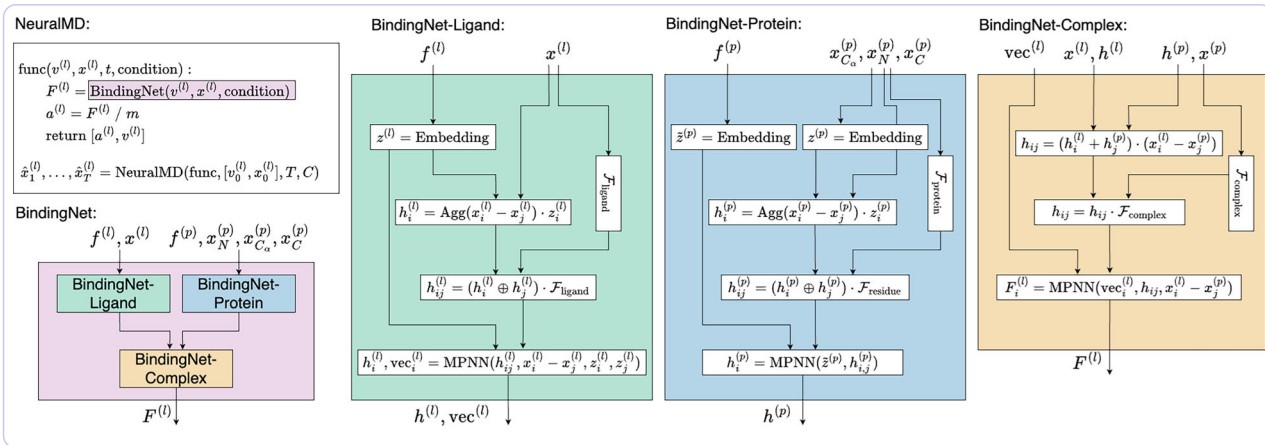

**Fig. 2 | The BindingNet architecture and integration. a** Illustration of the three granularities of vector frame bases utilized by BindingNet for geometric representation. **b** Overview of BindingNet's three core components and their integration into the NeuralMD pipeline for molecular dynamics simulation.

take both the mean absolute error (*MAE*) and root mean squared error (*RMSE*) between the predicted coordinates and ground-truth coordinates over all snapshots in the test set. (2) Validity. Molecules possess certain inherent properties, and we would like to measure the validity of the sampled positions. The first validity metric is the *Matching* metric. It is defined as the mean squared error between the atom pairwise distance from ground-truth conformations and our sampled conformations. This Matching metric reveals whether the model correctly reflects the true distance relationships in physical space. The second validity metric is Stability. It is defined as $\mathbb{P}_{i,j}(|\text{true distance} - \text{pred distance}| \leq \Delta)$, where we take $\Delta = 0.5$ Å. The underlying intuition is that, over long-time simulations, the predicted trajectory could enter an unrealistic state, such as bond breaking. Stability quantifies how often the predicted distances between atoms stay within a reasonable range of the true distances, indicating that the trajectory does not exhibit pathological behavior.

### Generalization Among Multiple Trajectories

The first task is to test the generalization ability of NeuralMD among different trajectories. The MISATO dataset includes 13,765 protein-ligand complexes[11], and we first create two small datasets by randomly sampling 100 and 1k complexes, respectively. Then, we take 80%-10%-10% for training, validation, and testing on both datasets. We also consider the whole MISATO dataset, where the data split has already been provided. After removing the complexes with peptide ligands, we have 13,066, 1357, and 1357 complexes for training, validation, and testing, respectively.

The quantitative results are in Table 1. The first observation is that VerletMD has worse performance on all three datasets, and the performance gap with other methods is even larger compared to the single-trajectory prediction, as will be introduced below. When comparing VerletMD, NeuralMD reaches up to 15 × reduction in reconstruction error and 70% increase in validity. The second observation is that the other two baselines, GNN-MD and DenoisingLD, show similar

performance, while NeuralMD outperforms in all datasets and all four metrics. The last observation is that the two validity metrics (Matching and Stability) are more distinguishable than the two trajectory reconstruction metrics (MAE and RMSE), suggesting that ligand oscillation may contain more informative details. We will explore this further in a qualitative study below.

### Generalization of one single trajectory

The second paradigm is the generalization of a single trajectory. The datasets are from the same protein-ligand trajectory, with the first 80 snapshots for training and the last 20 snapshots for testing.

The quantitative results are in Table 2. The first observation is that the baseline VertletMD has a clear performance gap compared to the other methods. There are two possible reasons: (1) Using ML models to predict the energy (or force) at each snapshot, and then using a numerical integration algorithm can fail in the long-time simulations; (2) ML for energy prediction methods require more data to train than the ML for coordinate prediction methods, thus they can perform worse in the low-data regime. The second observation is that, in general, both variants of NeuralMD demonstrate competitive performance across all ten tasks and four metrics, except for the reconstruction metrics on 3B9S and the validity metrics on 4K6W. The overall generalization performance reveals the potential of NeuralMD compared to existing ML baselines. Specifically, Stability (%) serves as a distinctive factor in method comparisons, with the two variants of NeuralMD outperforming on nine tasks by up to 70%.

### Qualitative analysis on oscillation

In addition to the quantitative measures like reconstruction and validity, we would also like to discuss the qualitative performance of ML models for MD simulation. We begin by analyzing the Root Mean Square Fluctuation of the ligand (**RMSF-Ligand**) along single trajectories. RMSF-Ligand quantifies the mean square deviation of each ligand atom's position from its average position over the trajectory,

**Table 1 | Results on three multi-trajectory binding dynamics predictions**

| Dataset | MISATO-100 | | | | MISATO-1000 | | | | MISATO-All | | | |
|---|---|---|---|---|---|---|---|---|---|---|---|---|
| | Reconstruction | | Validity | | Reconstruction | | Validity | | Reconstruction | | Validity | |
| | MAE | RMSE | Matching | Stability | MAE | RMSE | Matching | Stability | MAE | RMSE | Matching | Stability |
| VerletMD | 85.286 | 54.996 | 46.753 | 10.051 | 104.537 | 68.942 | 48.899 | 10.574 | 97.213 | 64.405 | 50.857 | 11.888 |
| GNN-MD | 5.964 | 3.938 | 0.671 | 70.546 | 7.524 | 4.915 | 0.670 | 68.310 | 7.637 | 5.048 | 0.675 | 69.244 |
| DenoisingLD | 8.251 | 5.541 | 1.744 | 29.545 | 9.251 | 6.074 | 1.362 | 37.289 | 8.149 | 5.387 | 0.764 | 68.315 |
| NeuralMD ODE (ours) | 5.867 | 3.870 | 0.539 | 79.553 | 7.459 | 4.867 | 0.612 | 70.362 | 7.513 | 4.961 | 0.491 | 81.991 |
| NeuralMD SDE (ours) | 5.868 | 3.871 | 0.533 | 80.229 | 7.476 | 4.876 | 0.457 | 83.960 | 7.517 | 4.963 | 0.474 | 83.264 |

Four evaluation metrics are considered: MAE (Å, ↓), RMSE (↓), Matching(↓), and Stability (%, ↑).

providing a measure of positional fluctuations. Formally, it is defined as:

$$\sqrt{\mathbb{E}_t\left[\frac{1}{N}\sum_{i=1}^{N}\| r_{i,t} - \langle r_i\rangle\|^2\right]}, \quad (2)$$

where $r_{i,t}$ is the $i$-th atom position at time $t$ (ground truth or sampled by ML methods) and $\langle r_i\rangle$ is the average position for the $i$-th atom across time. To capture localized oscillations more effectively, we employ a sliding window approach, calculating RMSF-Ligand over every five snapshots, such as snapshots [81, 85] and [96, 100].

RMSF-Ligand is a metric that measures the flexibility of individual atoms or residues during a simulation, reflecting how much they deviate from their average positions over time. It provides valuable insights into the extent of fluctuations in the system. Specifically, RMSF-Ligand describes the oscillations of individual trajectories. The objective here is to plot the RMSF-Ligand curves for the ground truth and four ML methods, enabling a comparison to determine which ML method aligns most closely with the ground truth curve. Notice that VerletMD is not considered as it cannot converge on the single-trajectory experiments (according to Table 2). The RMSF-Ligand curves are illustrated in Fig. 3. First, we observe that, overall, the two RMSF-Ligand curves produced by NeuralMD align better with the ground truth compared to those from ML baselines. Notably, NeuralMD demonstrates a significant improvement on datasets PDB 4ZX0, 3EOV, 4K6W, 4G3E, and 3B9S, and shows slightly better performance on 5WIJ, 1XP6, and 6B7F. Second, we note that almost all ML methods perform poorly on 1KTI and 4YUR. For these two trajectories, the ground truth exhibits sudden positional changes, representing out-of-distribution movements that remain an open challenge for the current ML community.

In addition to RMSF-Ligand, a more intuitive approach is to visualize the trajectories directly. We provide a 20-snapshot video comparing the trajectories generated by ML methods and the ground truth in the Supplementary Video 1. Additionally, we briefly present the sampled trajectories of four complexes using three ML methods alongside the ground truth in Fig. 3b. From these visualizations, it is evident that GNN-MD occasionally collapses, while DenoisingLD maintains a comparatively structured trajectory. Notably, NeuralMD demonstrates the highest stability across all cases.

## Efficiency Analysis

Efficiency is another important metric, and we consider measuring it with frames per second (FPS) on a single Nvidia-V100 GPU card. One main benefit of using NeuralMD for binding simulation is its efficiency. To show this, we list the computational time in Table 3. Since all ML methods share a similar backbone architecture (BindingNet and its variants), their efficiency differences arise from their computational approaches and hyperparameters. GNN-MD is the most efficient, as it requires no integration steps. NeuralMD incorporates an augmented

integration step, with the optimal step size set to 0.5 of the snapshot interval, i.e., two integration steps are performed to predict the next snapshot, making it slightly slower than VerletMD. In contrast, DenoisingLD is the slowest method, as it uses a smaller optimal time-step of one-tenth the snapshot interval, requiring 10 integration steps to generate the next snapshot during inference.

We further approximate the wall time using the numerical method for MD simulation (PDB 5WIJ). Concretely, we can get an estimated speed of 1 nanosecond of dynamics every 0.28 h. This is running the simulation with GROMACS[37] on 1 GPU with 16 CPU cores and a moderately sized water box at the all-atom level (around 64,000 atoms) with the stepsize of 2 femtoseconds. This shows that NeuralMD is approximately 25K × faster than numerical methods under optimal conditions. However, as noted earlier, the model's efficiency is highly sensitive to hyperparameter choices, so we consider an efficiency improvement factor of at least 1K to be a conservative estimate.

## Case Study on 4G3E: biological meaning of neuralMD

In this section, we provide a detailed discussion of PDB entry 4G3E, explore the potential advantages of using NeuralMD for drug development, and examine its biological implications.

Nuclear factor kappa B (NF-$\kappa$B) is a type of transcriptional factor that regulates transcription, cell survival, immune responses, and the development of the immune system[38-42]. NF-$\kappa$B inducing kinase (NIK), also known as MAP3K14, is a crucial signaling molecule that participates in the non-canonical NF-$\kappa$B pathway, which regulates immune responses and lymphoid organ development[40,42,43]. NIK is a serine/threonine kinase that activates the pathway by phosphorylating IKK$\alpha$ (I$\kappa$B kinase $\alpha$), which in turn leads to the processing of NF-$\kappa$B2 (p100) into its active form, p52. This activation allows the p52-RelB complex to translocate to the nucleus and initiate gene transcription that regulates various immune functions, including B cell survival, osteoclastogenesis, and lymphoid tissue development[42,43].

NIK is a promising drug target due to its pivotal role in the non-canonical NF-$\kappa$B signaling pathway[40,44-46]. Aberrant NIK activation and expression have been implicated in a range of pathological conditions, including autoimmune diseases, solid cancers, hematologic malignancies, cardiovascular disease, obesity, and type 2 diabetes[46]. Consequently, small-molecule inhibitors targeting NIK have demonstrated therapeutic potential by modulating immune responses and reducing inflammation, making NIK an attractive target for the treatment of various diseases[45,46].

Many drugs targeting NIK have been developed, with several structures of NIK bound to inhibitors resolved, providing valuable insights for drug design[40,47,48]. However, many existing drugs still require optimization to improve efficacy and specificity. Recent efforts used metabolite identification and structure-based drug design to improve pharmacokinetics, leading to inhibitors with reduced clearance and enhanced kinase selectivity, significantly lowering predicted human doses[45]. These advancements demonstrate how targeted optimization strategies can address key limitations. Our NeuralMD

**Table 2 | Results on ten single-trajectory binding dynamics predictions in the semi-flexible setting**

| PDB ID | Metric | VerletMD | GNN-MD | DenoisingLD | NeuralMD ODE (Ours) | NeuralMD SDE (Ours) |
|---|---|---|---|---|---|---|
| 5WIJ | MAE | 14.629 | 2.280 | 2.501 | 2.252 | 2.260 |
| | RMSE | 10.221 | 1.521 | 1.644 | 1.514 | 1.514 |
| | Matching | 5.459 | 0.803 | 0.815 | 0.464 | 0.615 |
| | Stability | 24.360 | 54.475 | 52.418 | 82.046 | 67.464 |
| 4ZX0 | MAE | 21.278 | 2.370 | 3.138 | 1.878 | 2.158 |
| | RMSE | 14.357 | 1.599 | 2.045 | 1.263 | 1.455 |
| | Matching | 7.971 | 0.555 | 1.072 | 0.428 | 0.696 |
| | Stability | 19.168 | 68.613 | 44.228 | 81.401 | 59.109 |
| 3EOV | MAE | 27.960 | 3.512 | 4.055 | 3.858 | 3.395 |
| | RMSE | 18.821 | 2.413 | 2.787 | 2.651 | 2.309 |
| | Matching | 13.588 | 1.216 | 1.209 | 1.062 | 0.962 |
| | Stability | 13.067 | 40.984 | 41.469 | 47.328 | 50.108 |
| 4K6W | MAE | 15.428 | 3.695 | 3.942 | 3.656 | 3.765 |
| | RMSE | 10.357 | 2.402 | 2.635 | 2.400 | 2.501 |
| | Matching | 7.505 | 1.038 | 0.839 | 0.928 | 1.076 |
| | Stability | 15.441 | 42.480 | 53.820 | 49.438 | 49.700 |
| 1KTI | MAE | 18.157 | 6.641 | 7.051 | 6.675 | 6.646 |
| | RMSE | 12.723 | 4.173 | 4.369 | 4.176 | 4.141 |
| | Matching | 7.467 | 0.386 | 0.268 | 0.337 | 0.167 |
| | Stability | 19.352 | 81.831 | 91.986 | 86.430 | 98.508 |
| 1XP6 | MAE | 13.753 | 2.378 | 2.218 | 1.924 | 2.061 |
| | RMSE | 9.587 | 1.561 | 1.472 | 1.280 | 1.356 |
| | Matching | 4.672 | 0.966 | 0.676 | 0.537 | 0.615 |
| | Stability | 28.129 | 49.239 | 64.951 | 75.533 | 69.423 |
| 4YUR | MAE | 16.764 | 7.031 | 7.128 | 6.957 | 7.038 |
| | RMSE | 11.069 | 4.641 | 4.807 | 4.597 | 4.679 |
| | Matching | 9.555 | 0.920 | 0.834 | 0.584 | 0.749 |
| | Stability | 16.542 | 47.555 | 49.676 | 69.775 | 60.344 |
| 4G3E | MAE | 5.111 | 2.709 | 3.588 | 2.191 | 2.345 |
| | RMSE | 3.503 | 1.785 | 2.321 | 1.453 | 1.536 |
| | Matching | 3.388 | 0.893 | 1.069 | 0.505 | 0.521 |
| | Stability | 31.852 | 61.802 | 40.823 | 71.436 | 68.729 |
| 6B7F | MAE | 31.934 | 4.136 | 4.431 | 3.921 | 3.842 |
| | RMSE | 22.168 | 2.768 | 3.047 | 2.652 | 2.601 |
| | Matching | 21.691 | 1.194 | 0.672 | 0.459 | 0.741 |
| | Stability | 11.050 | 39.067 | 61.583 | 75.692 | 57.917 |
| 3B9S | MAE | 19.473 | 2.578 | 2.811 | 3.039 | 3.132 |
| | RMSE | 11.696 | 1.699 | 1.868 | 1.999 | 2.078 |
| | Matching | 0.923 | 1.414 | 0.472 | 0.659 | 0.444 |
| | Stability | 57.801 | 49.306 | 71.852 | 76.065 | 77.801 |

Results with optimal training loss are reported. Four evaluation metrics are considered: MAE (Å, ↓), RMSE (↓), Matching(↓), and Stability (%, ↑).

approach can complement such strategies by offering dynamic insights into inhibitor binding, guiding the rational design of improved protein-targeted therapies.

The example of the complex structure of the inhibitor CMP1, a 6-alkynylindoline, bound to its target protein, NIK (Mus musculus; PDB: 4G3E), highlights the final binding state of the NIK inhibitor at its target region of NIK[40]. However, the dynamic process leading to this interaction remains poorly understood. Despite the development of numerous approaches for simulating these inhibitor binding processes (Fig. 3), their effectiveness is limited. For example, these methods, GNN-MD and DenoisingLD, frequently produce unrealistic structural distortions of CMP1 during the simulation (Fig. 3).

In contrast, our NeuralMD model not only accelerates the simulation of the binding trajectory between CMP1 and NIK but also avoids such exaggerated deformations. NeuralMD simulations

reveal that a specific structural region within the CMP1 molecule maintains stability or exhibits consistent, regular fluctuations during the binding process (Fig. 3b and Supplementary Video 1). These insights are invaluable for optimizing the NIK inhibitors, particularly by targeting molecular regions that enhance binding affinity and selectivity.

Furthermore, similar results have been observed across numerous examples in our studies (Supplementary Video 1). This demonstrates the broad applicability of NeuralMD for drug design, emphasizing its potential to streamline the development of therapeutics for various targets.

## Discussion

To sum up, we devise NeuralMD, an ML framework that incorporates a multi-grained group symmetric network architecture and second-

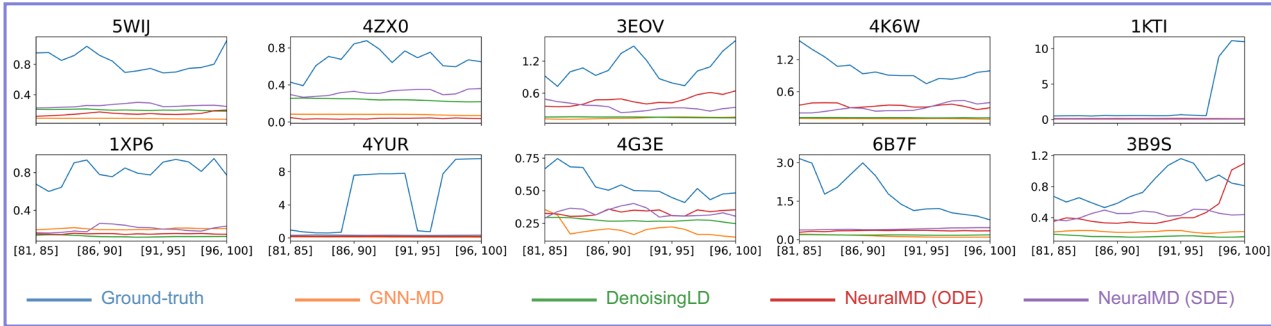

## (a) Visualization of RMSF-Ligand curves

## (b) Visualization of the last snapshot in protein-ligand dynamics

**Fig. 3 | Experiment results. a** Visualization of ligand oscillation via time-resolved root-mean-square fluctuation (RMSF) analysis. **b** Representative final snapshots of the four protein-ligand complexes from their respective molecular dynamics simulations. Source data are provided as a Source Data file.

**Table 3 | Efficiency comparison of FPS on single-trajectory prediction, with an accuracy of an integer**

| PDB ID | 5WIJ | 4ZX0 | 3EOV | 4K6W | 1KTI | 1XP6 | 4YUR | 4G3E | 6B7F | 3B9S | Average |
|---|---|---|---|---|---|---|---|---|---|---|---|
| VerletMD | 516 | 529 | 485 | 427 | 439 | 483 | 510 | 520 | 540 | 542 | 499.1 |
| GNN-MD | 830 | 806 | 763 | 715 | 756 | 768 | 823 | 810 | 823 | 844 | 793.8 |
| DenoisingLD | 291 | 299 | 285 | 285 | 269 | 299 | 296 | 298 | 299 | 300 | 292.1 |
| NeuralMD ODE | 429 | 434 | 424 | 393 | 361 | 423 | 430 | 433 | 440 | 434 | 420.1 |
| NeuralMD SDE | 440 | 407 | 411 | 396 | 368 | 424 | 434 | 429 | 441 | 429 | 417.9 |

order Newtonian mechanics, enabling accurate predictions of binding dynamics on a large timescale. Not only is such a timescale critical for understanding the dynamic nature of the ligand-protein complex, but our work marks the approach in developing a framework to predict coordinates for MD simulation in protein-ligand binding. We quantitatively and qualitatively verify that NeuralMD achieves superior performance on 13 binding dynamics tasks.

One potential limitation of our work is the dataset. Currently, we are using the MISATO dataset, a protein-ligand dynamics dataset with a large timestep. However, NeuralMD is agnostic to the timestep, and it

can also be applied to binding dynamics datasets with a timestep of a femtosecond. This may require the effort of the whole community for the dataset construction.

The second limitation of our work is GPU memory efficiency. Currently, we run multi- and single-trajectory MD simulations in a semi-flexible setting. While the flexible setting, where proteins can move, is more practical, it requires up to 100 times more GPU memory in the multi-trajectory case, as all protein conformations across 100 snapshots per trajectory must be modeled. For future work, we aim to develop a memory-efficient, physics-informed foundation

model for proteins, capable of rapidly adapting to MD simulation tasks, including MISATO.

Another limitation is the lack of verification through practical research. Although we have demonstrated the effectiveness of NeuralMD through computational simulations, we have not yet validated its biological relevance using experimental techniques, such as X-ray crystallization and cryogenic electron microscopy. While our model has shown promising results in terms of simulating protein-ligand interactions, its applicability in drug development requires further confirmation through practical research. To address this gap, we plan to synthesize the molecules based on the simulation results provided by NeuralMD. Such studies will help to enhance the reliability of NeuralMD as a tool for guiding the design of therapeutic agents.

## Methods

### Multi-grained vector frames

Proteins are macromolecules composed of up to thousands of residues (amino acids), where each residue is a small molecule. Thus, it is infeasible to model all the atoms in proteins due to the large volume of the system, and such an issue also holds for the protein-ligand complex. To address this issue, we propose BindingNet, a multi-grained SE(3)-equivariant model, to capture the interactions between a ligand and a protein. The vector frame basis ensures SE(3)-equivariance, and the multi-granularity is achieved by considering frames at three levels.

**Vector frame basis for SE(3)-equivariance.** Recall that the geometric representation of the whole molecular system needs to follow the physical properties of the equivariance w.r.t. rotation and translation. Such a group symmetric property is called SE(3)-equivariance. We also want to point out that the representation function should be reflection-equivariant for properties like energy, yet it is not for properties like chirality or ligand modeling in rigid protein structures. The vector frame basis inherently accommodates such reflection antisymmetry, and we leave a more detailed discussion in Supplementary C, along with the proof of group symmetry of the vector frame basis. In the following, we introduce three levels of vector frames for multi-grained modeling.

**Atom-level vector frame for ligands.** For small molecule ligands, we first extract atom pairs $(i, j)$ within the distance cutoff $c$, and the vector frame basis is constructed using the Gram-Schmidt as:

$$\mathcal{F}_{\text{ligand}} = \left( \frac{x_i^{(l)} - x_j^{(l)}}{\| x_i^{(l)} - x_j^{(l)} \|}, \frac{x_i^{(l)} \times x_j^{(l)}}{\| x_i^{(l)} \times x_j^{(l)} \|}, \frac{x_i^{(l)} - x_j^{(l)}}{\| x_i^{(l)} - x_j^{(l)} \|} \times \frac{x_i^{(l)} \times x_j^{(l)}}{\| x_i^{(l)} \times x_j^{(l)} \|} \right),$$
(3)

where $\times$ is the cross product. Note that both $x_i^{(l)}$ and $x_j^{(l)}$ are for geometries at time $t$ - henceforth, we omit the subscript $t$ for brevity. Such an atom-level vector frame allows us to do SE(3)-equivariant message passing to get the atom-level representation.

**Backbone-level vector frame for proteins.** Proteins can be treated as chains of residues, where each residue possesses a backbone structure. The backbone structure comprises an amino group, a carboxyl group, and an alpha carbon, delegated as $N - C_\alpha - C$. Such a structure serves as a natural way to build the vector frame. For each residue in the protein, the coordinates are $x_N$, $x_{C_\alpha}$, and $x_C$, then the backbone-level vector frame for this residue is:

$$\mathcal{F}_{\text{protein}} = \left( \frac{x_N - x_{C_\alpha}}{\| x_N - x_{C_\alpha} \|}, \frac{x_{C_\alpha} - x_C}{\| x_{C_\alpha} - x_C \|}, \frac{x_N - x_{C_\alpha}}{\| x_N - x_{C_\alpha} \|} \times \frac{x_{C_\alpha} - x_C}{\| x_{C_\alpha} - x_C \|} \right).$$
(4)

This is built for each residue, enabling the message passing for a residue-level representation.

**Residue-level vector frame for protein-ligand complexes.** It is essential to model the protein-ligand interaction to better capture the binding dynamics. We achieve this by introducing the residue-level vector frame. More concretely, proteins are sequences of residues, marked as

$$\left\{ (f_0^{(p)}, x_0^{(p)}), \ldots, (f_i^{(p)}, x_i^{(p)}), (f_{i+1}^{(p)}, x_{i+1}^{(p)}), \ldots \right\}$$

Here, we use a cutoff threshold $c$ to determine the interactions between ligands and proteins, and the interactive regions on proteins are called pockets. We construct the following vector frame on residues in the pockets sequentially:

$$\mathcal{F}_{\text{complex}} = \left( \frac{x_i^{(p)} - x_{i+1}^{(p)}}{\| x_i^{(p)} - x_{i+1}^{(p)} \|}, \frac{x_i^{(p)} \times x_{i+1}^{(p)}}{\| x_i^{(p)} \times x_{i+1}^{(p)} \|}, \frac{x_i^{(p)} - x_{i+1}^{(p)}}{\| x_i^{(p)} - x_{i+1}^{(p)} \|} \times \frac{x_i^{(p)} \times x_{i+1}^{(p)}}{\| x_i^{(p)} \times x_{i+1}^{(p)} \|} \right).$$
(5)

Through this vector frame, the message passing enables the exchange of information between atoms from ligands and residues from the pockets. The illustration of the above three levels of vector frames can be found in Fig. 2. Once we build up such three vector frames, we then design BindingNet, as will be introduced next. The key step involving vector frames is *scalarization* operation[49], which transforms the equivariant variables (e.g., coordinates) to invariant variables by projecting them to the three vector bases in the vector frame.

### Multi-Grained SE(3)-equivariant binding force modeling: bindingNet

In this section, we introduce BindingNet, a multi-grained SE(3)-equivariant geometric model for protein-ligand binding. The input of BindingNet is the geometry of the rigid protein and the ligand at time $t$, while the output is the force on each atom in the ligand.

**Atom-level ligand modeling.** We first generate the atom embedding using one-hot encoding and then aggregate each atom's embedding, $z^{(l)}$, by aggregating all its neighbors' embedding within the cutoff distance $c$. Then, we obtain the atom's equivariant representation by aggregating its neighborhood's messages as $(x_i^{(l)} - x_j^{(l)}) \cdot z_i^{(l)}$. A subsequent scalarization is carried out based on the atom-level vector frame as $h_{ij}^{(l)} = (h_i^{(l)} \oplus h_j^{(l)}) \cdot \mathcal{F}_{\text{ligand}}$, where $\oplus$ is the concatenation. Finally, it is passed through several equivariant message-passing layers (MPNN) defined as:

$$\text{vec}_i^{(l)} = \text{vec}_i^{(l)} + \text{Agg}_j \left( \text{vec}_i^{(l)} \cdot \text{MLP}(h_{ij}) + (x_i^{(l)} - x_j^{(p)}) \cdot \text{MLP}(h_{ij}) \right), \quad (6)$$

where MLP($\cdot$) and Agg($\cdot$) are the multi-layer perceptron and mean aggregation functions, respectively. vec $\in \mathbb{R}^3$ is a vector assigned to each atom and is initialized as 0. The outputs are atom representation and vector ($h^{(l)}$ and vec$^{(l)}$), and they are passed to the complex module introduced below.

**Backbone-level protein modeling.** For the coarse-grained modeling of proteins, we consider three backbone atoms in each residue. We first obtain the atom embedding on three atom types, and then we obtain each atom's representation $z^{(p)}$ by aggregating its neighbor's representation. Then, we obtain an equivariant atom representation by aggregating the edge information, $(x_i^{(p)} - x_j^{(p)}) \cdot z_i^{(p)}$, within cutoff distance $c$. Following which is the scalarization on the residue frame $h_{ij}^{(p)} = (h_i^{(p)} \oplus h_j^{(p)}) \cdot \mathcal{F}_{\text{protein}}$. Recall that we also have the residue type, and with a type embedding $\tilde{z}^{(p)}$, we can obtain the final residue-level

representation using an MPNN layer as:

$$h^{(p)} = \tilde{z}^{(p)} + (h_{N,C_\alpha}^{(p)} + h_{C_\alpha,C}^{(p)})/2. \qquad (7)$$

**Residue-Level Complex Modeling.** Once we obtain the atom-level representation and vector $(h^{(l)}, \text{vec}^{(l)})$ from ligands, and backbone-level representation $(h^{(p)})$ from proteins, the next step is to learn the protein-ligand interaction. We first extract the residue-atom pair $(i,j)$ with a cutoff $c$, based on which we obtain an equivariant interaction edge representation $h_{ij} = (h_i^{(l)} + h_j^{(p)}) \cdot (x_i^{(l)} - x_j^{(p)})$. After scalarization, we can obtain invariant interaction edge representation $h_{ij} = h_{ij} \cdot \mathcal{F}_{\text{complex}}$. Finally, we adopt an equivariant MPNN layer to get the atom-level force as:

$$\text{vec}_{ij}^{(pl)} = \text{vec}_i^{(l)} \cdot \text{MLP}(h_{ij}) + (x_i^{(l)} - x_j^{(p)}) \cdot \text{MLP}(h_{ij}). \qquad (8)$$

The ultimate force predictions for each atom include two parts: the internal force from the molecule $\text{vec}_i^{(l)}$ and the external force from the protein-ligand interaction $\text{vec}_{ij}^{(pl)}$. To sum them up, we have:

$$F_i^{(l)} = \text{vec}_i^{(l)} + \text{Agg}_{j \in \mathcal{N}(i)} \text{vec}_{ij}^{(pl)}. \qquad (9)$$

These three modules consist BindingNet. More complete descriptions can be found in Supplementary E and Supplementary Fig. S3.

### Binding MD modeling: neuralMD

As previously clarified, MD follows Newtonian mechanics, and we solve it as either an ODE problem or an SDE problem. The BindingNet introduced above takes in the molecular system geometry at time $t$ and outputs the forces. Then in this section, we describe how we use the neural differential equation solver to predict the coordinates at future snapshots.

We want to highlight that one ML for the MD simulation research line is predicting the energy or force[9,31,32], which will be fed into the numerical integration algorithms for trajectory simulation. For accuracy, such an ML-based MD simulation must be at the femtosecond level (1e-15 s). However, as shown in recent works[21,50,51], minor errors in the ML force field can lead to catastrophic failure for long-time simulations. For instance, there can be pathological behavior, such as extreme force predictions or bond breaking, within the low end of the distribution. Our experiments have yielded similar observations, as will be shown below. In this paper, however, we overcome this issue by directly learning the extended-timescale MD trajectories (nanosecond level, 1e-9 s).

### Newtonian dynamics and Langevin dynamics for MD simulation.

For MD simulation, if we assume the information of all particles in the molecular system (e.g., solvent molecules) is known and no thermal fluctuation is considered, then it can be modeled as Newtonian dynamics, which is essentially an ODE. For modeling, we consider using the BindingNet introduced above for force prediction at time $\tau$, as:

$$F_\tau^{(l)} - \text{ODE} = \text{BindingNet}(f^{(l)}, x_\tau^{(l)}, f^{(p)}, x_N^{(p)}, x_{C_\alpha}^{(p)}, x_C^{(p)}). \qquad (10)$$

On the other hand, Langevin dynamics introduces a stochastic component for large molecular systems with thermal fluctuations. Concretely, Langevin dynamics is an extension of the standard Newtonian dynamics with the addition of damping and random noise terms: $F - \gamma m v + \sqrt{2m\gamma k_B T} R(t)$, where $\gamma$ is the damping constant or collision frequency, $T$ is the temperature, $k_B$ is the Boltzmann's constant, and $R(t)$ is a delta-correlated stationary Gaussian process with zero-mean. For our implementation, we learn this equation and arguments in a data-driven way with reparameterization as:

$$\begin{aligned} F_\tau^{(l)}\text{-SDE} = &\text{BindingNet}(f^{(l)}, x_\tau^{(l)}, f^{(p)}, x_N^{(p)}, x_{C_\alpha}^{(p)}, x_C^{(p)}) \\ &+ \text{BindingNet-Ligand}(f^{(l)}, x_\tau^{(l)}) \cdot \epsilon, \end{aligned} \qquad (11)$$

where $\epsilon \sim \mathcal{N}(0,1)$ is the standard Gaussian noise. Ultimately, adopting either $F_\tau^{(l)}$-ODE or $F_\tau^{(l)}$-SDE as the force prediction $F_\tau^{(l)}$ on each atom, the coordinates at time $t$ can be obtained after integration as:

$$a_\tau^{(l)} = \frac{F_\tau^{(l)}}{m}, \qquad \hat{v}_t^{(l)} = v_0^{(l)} + \int_0^t a_\tau^{(l)} d\tau, \qquad \hat{x}_t^{(l)} = x_0^{(l)} + \int_0^t \hat{v}_\tau^{(l)} d\tau. \qquad (12)$$

The training objective is the mean absolute error (MAE) or root mean squared error (RMSE) between the predicted coordinates and ground-truth coordinates along the whole trajectories:

$$\mathcal{L} = \mathbb{E}_t \left[ ||\hat{x}_t^{(l)} - x_t^{(l)}|| \right]. \qquad (13)$$

An illustration of NeuralMD pipeline is in Fig. 2.

### Second-order differential equation.

We also want to highlight that NeuralMD is solving a second-order differential equation, as in Newtonian and Langevin dynamics. The key module of the neural differential method is the differential function[23], which returns the first-order derivative. Then, the outputs of the differential function will be integrated using algorithms like the Euler algorithm. To learn the MD trajectory following second-order ODE and SDE, we propose the following formulation of the second-order equation within one integration call:

$$\begin{bmatrix} dx/dt \\ dv/dt \end{bmatrix} = \begin{bmatrix} v \\ F/m \end{bmatrix}. \qquad (14)$$

We mark this as "func" in Fig. 2. This means we can augment ODE or SDE derivative space by concurrently calculating the accelerations and velocities, allowing simultaneous integration of velocities and positions.

### Implementation details on velocity modeling.

As in Eq. (12) and (14), NeuralMD requires both initial coordinates and velocities to predict the trajectories of coordinates and velocities. However, the MIS-ATO dataset we are using does not include velocity information. To handle this issue, we introduce two solutions: (1) We first estimate the initial velocity as the positional momentum, $v_t^{(l)} = (x_t^{(l)} - x_{t-1}^{(l)})$. Optionally, we can apply a velocity initial mapping function to get a surrogate initial velocity: $v_t^{(l)} = \text{BindingNet-Ligand}(f^{(l)}, x_t^{(l)} - x_{t-1}^{(l)})$. The decision to include a velocity mapping is treated as a binary hyperparameter. (2) During the integration step in (14), we can also take an extra module to refine the velocity. More specifically, we have $v_t^{(l)} = v_t^{(l)} + \alpha \cdot \text{BindingNet-Ligand}(f^{(l)}, x_t^{(l)})$, where $\alpha$ is the velocity refinement coefficient. This is not displayed from Fig. 2 for brevity, yet all critical details are provided to ensure reproducibility.

### Reporting summary

Further information on research design is available in the Nature Portfolio Reporting Summary linked to this article.

## Data availability

The data utilized in this work is available at MISATO[11]. Detailed instructions for downloading and preprocessing the dataset are provided at this GitHub repository. The source data underlying Fig. 3, Supplementary Figs. S1-S2, and Supplementary Tables S3-S6 are provided as a Source Data file. Source data are provided with this paper.

## Code availability

The code is available on this GitHub repository and Zenodo[52]. For the figures generated in Figs. 1 and 3, we are using the PyMol tool.

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

## Acknowledgements

The authors would like to thank the financial support from the Bakar Institute of Digital Materials for the Planet (BIDMaP).

## Author contributions

All authors contributed to the project discussion and paper writing. S.L., W.D., H.X., Y.L., Z.L., V.B., C.B., A.A., H.G., and J.C. conceived and designed the experiments. S.L., W.D., and Y.L. performed the experiments. S.L., W.D., H.X., Y.L., Z.L., V.B., and D.Y. analyzed the data. S.L., H.X., Y.L., Z.L., V.B., and D.Y. contributed analysis tools.

## Competing interests

The authors declare no competing interests.
