## [Transparent Peer Review file · Nature Communications]

A Multi-Grained Symmetric Differential Equation Model for Learning Protein-Ligand Binding Dynamics

Corresponding Author: Dr Shengchao Liu

Version 0:

Reviewer comments:

Reviewer #1

(Remarks to the Author)

This paper introduces a machine learning approach for molecular dynamics (MD) simulations of protein-ligand complexes. Specifically, the proposed method incorporates two main components: a multi-scale SE(3)-equivariant neural network and two different neural differential equation solvers inspired by Newton's second law and Langevin dynamics. The method is validated on the MISATO dataset, with a custom set of baselines and evaluation metrics demonstrating its advantages in both speed and accuracy. However, the reviewers raised several concerns regarding the experimental setup, choice of baselines, methodology, and experimental outcomes:

1. Baselines

The study evaluates the proposed method against only three deep learning-based baselines, ensuring a fair comparison by using the same backbone across different methods. However, could the authors include other MD methods, even those with different types of backbones, to present more competitive results? While the paper highlights a 1000x speed improvement over standard numerical MD, it lacks direct comparisons of both accuracy and speed with traditional computational MD approaches.

2. Model Comparisons and Ablation Studies:

While the method employs multi-scale modeling of protein-ligand complexes, many other approaches exist for handling equivariant complex inputs. Comparisons with alternative network types, such as those developed for docking[1,2,3] or structure-based drug design (SBDD)[4] scenarios, would strengthen the study. Additionally, ablation studies are absent, making it difficult to assess the contributions of different network components or the multi-scale design to the task.

[1] EquiBind: Geometric Deep Learning for Drug Binding Structure Prediction

[2] FABind: Fast and Accurate Protein-Ligand Binding

[3] DiffDock-Pocket: Diffusion for Pocket-Level Docking with Side Chain Flexibility

[4] MolCRAFT: Structure-Based Drug Design in Continuous Parameter Space

3. Task Design and Protein Invariance Assumption

The evaluation metrics primarily focus on the accuracy of position and distance, without addressing binding affinity or energy-related measures. This makes it challenging to understand how the simulation results directly contribute to drug discovery applications. The paper also assumes the protein structure to be invariant, which might oversimplify the dynamics and deviate from real-world MD scenarios involving protein-ligand complexes.

Furthermore, the claim that "all prior ML approaches for MD simulation are limited to single-system and not protein-ligand complex" is not appropriate. In fact, AI2BMD [5] is a ML method to consider complex and consider both protein and ligand changes in the MD process, which has been published in Nature. More discussion about the differences and benefits of the proposed method against AI2BMD should be included.

[5] Ab initio characterization of protein molecular dynamics with AI2BMD

4. Case Study on 4G3E

While NIK is an interesting target, the case study lacks in-depth analysis. Additional results are necessary to demonstrate how the proposed method enhances drug binding affinity and selectivity. Experimental validation (e.g., wet-lab experiments) or highly accurate computational methods (e.g., FEP, TI) should be included.

5. About Simultaneous Calculation of Acceleration and Velocity

The paper mentions:

"We augment derivative space by concurrently calculating the accelerations and velocities, allowing simultaneous integration of velocities and positions."

However, it does not provide sufficient algorithmic details. Numerical integration of positions requires velocity values at corresponding time steps, while velocity calculations require prior integration of accelerations. Please clarify how "simultaneous" calculations are achieved and describe the numerical integration method, including the choice of discretization step size.

6. About Efficiency

Why does the NeuralMD algorithm reduce the need for discretization compared to the DenoisingLD method, enabling conformational prediction with fewer time steps? Please provide a detailed explanation.

7. Pretraining

Can pretraining on data annotated with energy/force labels further improve the performance?

8. Suggestions for Improving Figures and Names

The information provided in Figure 1 is limited, as it only presents the formula for the Velocity Verlet algorithm. It is recommended to enhance the figure by including additional information, such as comparisons with GNN-MD and DenoisingLD. This would help emphasize the innovative aspects of the proposed method. The design and color scheme of Figure 2a could be improved for better aesthetics and clarity.

Besides, I suggest the authors consider using another name of BindingNet. Because it has been used by a protein-ligand complex data, which may easily cause confusions.

(Remarks on code availability)

Reviewer #2

(Remarks to the Author)

The work by Liu et al. represents ongoing efforts to expedite MD and related computational methods for drug discovery applications using ML. It seems that they achieved significant improvement on efficiency of MD simulations. However, this work lacks a few things for it to be useful in a drug discovery campaign.

1. It is not clear that with the NeuralMD, one can perform interaction analysis throughout trajectories.
2. As the authors point out, proteins being fixed during the simulation is a major drawback, as one usually resorts to MD simulations for exactly the cases in which protein structures are flexible. This assumption severely limits the utility of the method in drug discovery.
3. Training dataset is in the order of 10,000, which doesn't seem to be enough. I wonder if one can augment it by generating protein-ligand complex structures by docking.

I would like to see how these issues are addressed before recommending the manuscript for publication.

(Remarks on code availability)

Reviewer #3

(Remarks to the Author)

In this study, the authors developed a multi-grained symmetric differential equation model for learning protein-ligand binding dynamics. The motivation of this work is that the conventional molecular dynamics method is too slow and too computationally expensive for predicting protein-ligand structures. The method, NeuralMD, outperforms other ML approaches, such as GNN-MD, DenoisingLD, etc, as for the accuracy, stability, and performances. The results look good due to the comparisons. However, for this reviewer, who is not an expert of the ML field, there are many fundamental questions about the manuscript in the current form.

1) Purpose of this work

What is the main purpose of this work? Do they want to predict the binding processes or just the binding poses? Since they often use the word, "trajectories", like MD simulations, I was confused for their main research purposes.

2) Initial structures

According to the movies on their github, the motions of ligand are relatively small. Though they are flexible, their positions relative to the protein do not change so much. What are the initial structures in the predictions?

If they separated a protein and a ligand, how they can predict the protein-ligand complexes?

What are their target conformational spaces to be sampled by ML?

3) Binder and non-binder distinctions

Can they distinct binder ligands or non-binder ligands in their methods?

4) RMSF in Figure 3

What is the ground-truth RMSF? Is it MD simulation result or B-factor in X-ray structure?

Why the RMSF in ML is generally so small?

What is the physical meaning of the RMSF?

Why the ground-truth RMSF in 1KTI is so large?

These questions are fundamental and useful to increase the readability of this manuscript to non-expert chemists. I want to hear the authors' replies.

(Remarks on code availability)

Version 1:

Reviewer comments:

Reviewer #1

(Remarks to the Author)

I appreciate the author's revisions to the paper and figures, as well as the renaming of the method. My questions regarding the distinction between energy/force and MD tasks, the explanation of pretraining, and the subsequent experimental validation plan have been resolved.

I also thank the author for the explanations regarding SE(3) equivariance, E(3) equivariance, and reflection-antisymmetry in the network. However, even if these methods don't explicitly emphasize the concepts of SE(3) equivariance and reflection-antisymmetry, they also focus on multi-scale modeling of protein-ligand interactions, which is a central aspect of this work. Without a comparison, it's difficult to assess the practical significance of the proposed network's architectural innovations.

Therefore, I encourage the authors to conduct such comparisons and provide corresponding discussions or empirical analyses to better highlight the novelty and effectiveness of their design.

(Remarks on code availability)

Reviewer #2

(Remarks to the Author)

As the authors admit, there are limitations to their proposed method. However, with their detailed response, I am convinced that NeuralMD has utility in the current drug discovery practice, though there is quite a bit of room to improve. I recommend the publication of the manuscript.

(Remarks on code availability)

Reviewer #3

(Remarks to the Author)

In the discussion with reviewers, the limitation of this machine learning model becomes clearer. The rigid treatment of protein is a significant approximation compared to the flexible molecular dynamics simulations. In this sense, their statement about the comparison between the ML surrogate and MD is indeed an overstatement. For instance, in the abstract, they wrote:

We demonstrate the efficiency and effectiveness of NeuralMD, achieving over 1K× speedup compared to standard numerical MD simulations.

But, considering the rigid treatment in the ML surrogate and the flexible treatment in the MD simulation, I cannot accept the statement. The author needs to emphasize the limitation of the current model in the abstract and maintext and discuss the future perspectives about the flexible treatment of protein structures.

Also, the physical meaning of RMSF shown in Figure 3 is different from the RMSF in MD simulations, because of the different treatment of protein flexibility. If such quantities are shown without reasonable explanations, the data will confuse readers of this manuscript.

Also, the treatment of solvent is different between the ML surrogate model and the MD simulation. In the MD simulation, solvent molecules are explicitly included, while the ML surrogate does not.

I suggest that the author compared these two different methods with the same conditions and approximations. Otherwise, just the numerical comparison of the computational time is not meaningful.

(Remarks on code availability)

Version 2:

Reviewer comments:

Reviewer #3

(Remarks to the Author)

The authors have improved the manuscript significantly.

(Remarks on code availability)

Revisions made to manuscript NCOMMS-24-74647

We would like to extend our sincere appreciation to the editor and reviewers for their invaluable comments and suggestions, which have significantly enhanced the quality of this manuscript. In response, we want to highlight four key points:

1. In this community, many papers have been using $E(3)$ -equivariant networks for modeling the molecule geometry. However, they missed the key difference between $E(3)$ -equivariance and $SE(3)$ -equivariance, especially reflection anti-symmetry, which is very important in the binding tasks. We highlighted how we handled this using the three granularities of vector frames in our NeuralMD pipeline, in Sections 4, B, and C (with detailed proof).
2. We highlighted that there are two different research lines in AI for MD simulation, one is **ML for Potential Energy and Force Learning for MD Simulation**, and the other one is **ML for Trajectory Learning for MD Simulation**. We listed the details and comparison in Section A. Note that our current version of NeuralMD belongs to the second category.
3. As discussed in Section 3, we currently adopt a semi-flexible setting due to computational limitations. This setting has also been used in prior works [1, 2]. While we acknowledge that this is not a perfect solution, we view NeuralMD as an initial step in this direction. Our group is actively pursuing follow-up work to further advance this line of research.
4. Meanwhile, NeuralMD still supports primary interaction analysis, as demonstrated in the 4G3E case study in Section 2.8 and the 10 binding complex examples in the supplementary files (also on the website). These observations suggest that NeuralMD is a promising direction for future exploration, especially from an AI and physics perspective.

To summarize, while we acknowledge the current limitations of NeuralMD, we believe it addresses important challenges in the community and offers a valuable step forward. We are grateful for the constructive feedback, and we have meticulously revised the manuscript (marked in green) in accordance with the insightful comments. We would also be grateful if they could recognize the contributions of our proposed multi-grained vector frame modeling approach. Please find our detailed responses below.

Reviewer 1

Thank you for raising these detailed and insightful questions about our work. We have carefully addressed each point with additional explanations as outlined below.

Reviewer Comment 1.1 — Baselines. The study evaluates the proposed method against only three deep learning-based baselines, ensuring a fair comparison by using the same backbone across different methods. However, could the authors include other MD methods, even those with different types of backbones, to present more competitive results? While the paper highlights a 1000x speed improvement over standard numerical MD, it lacks direct comparisons of both accuracy and speed with traditional computational MD approaches.

Reply: Thank you for raising these questions. We would like to answer the key points below.

- First would like to point out that this is a new research direction, on **ML for trajectory learning for MD simulation**, as summarized in Section A.
- In our work, the 1000 \times speedup is measured relative to the traditional numerical MD method, specifically GROMACS. This reflects the speed comparison. Meanwhile, the MISATO dataset [3] provides the binding dynamics data used in our experiments, which were generated using these numerical MD simulations. In other words, while we achieve significant acceleration, the ground-truth trajectories are obtained from accurate numerical methods.
- By ‘... other MD methods, even those with different types of backbones...’ Do you mean other AI-MD methods? If so, then it is related to Comment 1.2, and we would like to discuss the details below accordingly.

Reviewer Comment 1.2 — Model Comparisons and Ablation Studies: While the method employs multi-scale modeling of protein-ligand complexes, many other approaches exist for handling equivariant complex inputs. Comparisons with alternative network types, such as those developed for docking[1,2,3] or structure-based drug design (SBDD)[4] scenarios, would strengthen the study. Additionally, ablation studies are absent, making it difficult to assess the contributions of different network components or the multi-scale design to the task.

[1] EquiBind: Geometric Deep Learning for Drug Binding Structure Prediction [2] FABind: Fast and Accurate Protein-Ligand Binding [3] DiffDock-Pocket: Diffusion for Pocket-Level Docking with Side Chain Flexibility [4] MolCRAFT: Structure-Based Drug Design in Continuous Parameter Space

Reply:

Thank you for mentioning these works.

Firstly, we would like to acknowledge that we are aware of this line of research, including [1, 2, 4]. However, these works are not directly applicable to our setting due to a fundamental mathematical issue: They employ E(3)-equivariant networks, following EGNN [4], which are not suitable for binding tasks which require the model to be E(3)-equivariant and reflection-antisymmetry. We provide a

Figure 1: Explanation of SE(3)-equivariance, E(3)-equivariance, and SE(3)-equivariance and reflection-antisymmetry.

detailed explanation of this limitation in the paragraph **SE(3)-equivariance, E(3)-equivariance, and reflection-antisymmetry** below.

- EquiBind, E3-equivariant in Section 3, link.
- FABind, E3-equivariant in Section 3.2, link. Besides, this is a parallel work.
- MolCRAFT, E3-equivariant in Eq 25, link. Besides, this is also a parallel work.

Secondly, from a modeling perspective, our approach—based on SE(3)-equivariance and reflection-antisymmetry—can be seen as an extension of the vector frame concept introduced in AlphaFold2 [5]. In AlphaFold2, the authors construct a local frame for each residue using three backbone atoms. We extend this idea by designing vector frames at three levels of granularity, as shown in Figure 2a of the manuscript: a ligand frame, a protein frame, and a complex frame.

Continuing the discussion on geometric modeling, we believe there are three reasons why adding the work [3] as a baseline is not necessary: (1) The DiffDock-Pocket paper uses the Tensor Field Network (TFN) [6], which was the first architecture to incorporate SE(3)-equivariance and reflection-antisymmetry. However, more expressive and effective approaches have since been developed. (2) In recent top-tier ML conferences, vector frame-based modeling is the state-of-the-art for protein geometric modeling, as demonstrated by works such as FrameDiff [7], FrameFlow [8], FoldFlow [9], and FoldFlow2 [10]. (3) DiffDock-Pocket is a parallel work that has not yet been peer-reviewed.

SE(3)-equivariance, E(3)-equivariance, and SE(3)-equivariance and reflection-antisymmetry.

SE(3)-equivariance encompasses translation and rotation in Euclidean space but does not account for reflection or chirality equivariance. Then, as shown in Figure 1, there are two cases:

- SE(3)-equivariance and reflection-symmetry, or equivalently E(3)-equivariance. In the EGNN paper [4], the authors explicitly state: “It is often desired that predictions on these tasks are either equivariant or invariant w.r.t. E(3) transformations.” However, the paper does not provide a qualitative or quantitative assessment of how *often* this assumption holds in practice.
- SE(3)-equivariance and reflection-antisymmetry. This stands in contrast to E(3)-equivariance, and there are two reasons that we should stick to this when applying the geometric modeling.
 1. In the supplementary information of the AlphaFold2 paper [5] – specifically, Supplementary Figure 9 (link) – the authors demonstrate that natural molecules are not reflection-symmetric, but rather exhibit reflection-antisymmetry.

- When modeling protein–ligand binding tasks, it is important to incorporate both SE(3)-equivariance and reflection-antisymmetry for the protein and ligand. Using E(3)-equivariance instead can lead to undesirable behavior. For example, if the protein is fixed and the small molecule is reflected, an E(3)-equivariant model would produce reflected predictions—such as forces—which is physically incorrect and inconsistent with the nature of molecular interactions.

Besides, in our supplementary information, we explicitly explain how we can guarantee the SE(3)-equivariance and reflection-antisymmetry in Section C. Please feel free to check if you are interested.

Reviewer Comment 1.3 — 3. Task Design and Protein Invariance Assumption The evaluation metrics primarily focus on the accuracy of position and distance, without addressing binding affinity or energy-related measures. This makes it challenging to understand how the simulation results directly contribute to drug discovery applications. The paper also assumes the protein structure to be invariant, which might oversimplify the dynamics and deviate from real-world MD scenarios involving protein-ligand complexes. Furthermore, the claim that "all prior ML approaches for MD simulation are limited to single-system and not protein-ligand complex" is not appropriate. In fact, AI2BMD [5] is a ML method to consider complex and consider both protein and ligand changes in the MD process, which has been published in Nature. More discussion about the differences and benefits of the proposed method against AI2BMD should be included.
[5] Ab initio characterization of protein molecular dynamics with AI2BMD

Reply: Thank you for raising the questions on the problem setting, energy, and related works.

Problem formulation. First of all, we would like to mention that the problem formulation is:

$$\mathcal{L} = \mathbb{E}_t [|\hat{\mathbf{x}}_t^{(l)} - \mathbf{x}_t^{(l)}|]. \quad (1)$$

BTW. This is Equation 13 in the manuscript. We would like to point out a few key differences or attributes of our problem formulation:

- This is different from the other research line on energy and force prediction for MD simulation.
- There are two key differences between energy prediction for MD and trajectory prediction for MD: (1) Energy prediction takes each conformation and energy as IID, while trajectory learning optimizes the conformations along the whole trajectory, enforcing the temporal relation. (2) The trajectory learning is agnostic to the magnitude of the timesteps, and energy prediction can be sensitive to longer-timestep MD simulations.
- This research line includes CGDMS [11], DiffMD [12], DFF [13], CG-MD [14].
- We listed the detailed comparison in Section A.2: the comparison between **ML for Potential Energy and Force Learning for MD Simulation** and **ML for Trajectory Learning for MD Simulation**.

About the problem formulation of the semi-flexible setting. Assuming the protein stays fixed because:

- This is due to the large volume of proteins.
- Modeling both the ligand and protein and applying the second-order integration for simulating the dynamics goes above the GPU memory limitation.

- This assumption is also common in the top-tier community, *e.g.*, DiffDock [1]. BTW, we also found that the DiffDock-PP [2] shared by you using the inflexible setting, and they claimed that “This assumption is often realistic (Vakser, 2014) and even leads to improved results for most interacting proteins (Desta et al., 2020).”

Then on the analysis. We acknowledge that NeuralMD has certain limitations, as we discussed in Section 3 of the manuscript, including GPU limitations for the flexible setting. However, NeuralMD still possesses the ability for certain primary interaction analysis.

- We provide a case study on 4G3E, as described in the manuscript Section 2.8, as well as in Comment 1.4.
- We provide the demonstration for 10 binding complex examples in the supplementary files (also on the website).

These observations reveal that NeuralMD is a promising direction for future exploration, especially from the AI and physics perspective.

About the related works. We would like to kindly point out that the first version of our work was released in September 2023, and AI2BMD is indeed a parallel work.

Besides, we also checked if AI2BMD open-sourced the codes, and the authors stated on Jan 14th 2025 that “The current version support single chain protein simulations, while more functions will be added and released later”. Please check this link for more details.

Meanwhile, we would also like to point out, as will be discussed in Comment 1.7, the problem formulation of AI2BMD is **ML for Potential Energy and Force Learning for MD Simulation**, while ours is in the **ML for Trejectory Learning for MD Simulation** research line. There are two key differences between energy and trajectory prediction for MD (we also discussed them in the comments above): (1) Energy prediction takes each conformation and energy as IID, while trajectory learning optimizes the conformations along the whole trajectory, enforcing the temporal relation. (2) The trajectory learning is agnostic to the magnitude of the timesteps, and energy prediction can be sensitive to longer-timestep MD simulations.

Please feel free to check more discussions in Section A.2. BTW. We have added this discussion to the revised manuscript in Section A.2.

Reviewer Comment 1.4 — Case Study on 4G3E. While NIK is an interesting target, the case study lacks in-depth analysis. Additional results are necessary to demonstrate how the proposed method enhances drug binding affinity and selectivity. Experimental validation (*e.g.*, wet-lab experiments) or highly accurate computational methods (*e.g.*, FEP, TI) should be included.

Reply: Thank you for your valuable feedback.

We agree that NIK is an important kinase involved in the regulation of immunity and inflammation, and it has also been identified as a potential therapeutic target in glioma, melanoma, and prostate cancer [15, 16]. Your suggestion to perform wet-lab validation or employ highly accurate computational methods such as Free Energy Perturbation (FEP) and Thermodynamic Integration (TI) is indeed a standard and rigorous approach for confirming improvements in the drug binding affinity and selectivity as simulated by our NeuralMD model. However, we plan to continue this work in a future project. Our

decision is based on the following two reasons:

- **The application of FEP/TI does not provide stronger evidence for demonstrating the acceleration of molecular dynamics (MD) simulations achieved by our model compared to what is shown in Figure 3, and the current architecture of our model is not yet well-suited for reliably executing FEP/TI simulations.** We agree that FEP and TI are powerful tools for comparing the relative binding affinities of different ligands to the same protein target. In our case, this would require identifying additional molecules that bind to NIK for comparison with CMP1, a 6-alkynylindoline. However, the primary focus of this work is to introduce our NeuralMD model, which achieves over a 1000-fold improvement in MD simulation speed across various protein–ligand systems (Figure 3). We believe this demonstration provides stronger evidence of the model’s capability in accelerating MD simulations than a comparison of the relative binding free energies of different ligands for a single target. Moreover, our current model architecture is not well-suited to support the execution of FEP/TI simulations. However, we recognize the value of such comparative studies and plan to include them in future work. We are currently applying the model to novel molecules not present in the MISATO dataset, and extending the model to integrate FEP/TI represents a promising direction to further assess both MD simulation speedup and accuracy against known protein targets such as NIK.
- **The experimental validation results related to NIK have already been discussed in our paper, and additional wet-lab experiments require further resources.** The binding between CMP1 and NIK has already been experimentally validated in the original publication that reports the crystal structure 4G3E [17]. The structure itself is the result of wet-lab crystallographic work, and has been discussed in Section 2.8. We are currently in contact with wet-lab collaborators to screen and evaluate additional molecules that bind to NIK, in parallel with MD simulation using our NeuralMD model.

The main goal of our paper is to introduce our NeuralMD model, which can be applied across various protein–ligand systems and offers over a 1000-fold speedup in MD simulations. Therefore, we did not devote extensive space to NIK-specific analysis. However, we sincerely appreciate your thoughtful suggestion and fully intend to incorporate both highly accurate computational validations and wet-lab experiments in our future research.

Reviewer Comment 1.5 — 5. About Simultaneous Calculation of Acceleration and Velocity
The paper mentions: "We augment derivative space by concurrently calculating the accelerations and velocities, allowing simultaneous integration of velocities and positions." However, it does not provide sufficient algorithmic details. Numerical integration of positions requires velocity values at corresponding time steps, while velocity calculations require prior integration of accelerations. Please clarify how "simultaneous" calculations are achieved and describe the numerical integration method, including the choice of discretization step size.

Reply: Thank you for raising this question.

This formulation is indeed rooted in physics, as presented in Equation 14 in the manuscript. We are happy to provide additional details here.

So the goal is to calculate the second-order integration, as Equation 12 in the manuscript:

$$\mathbf{a}_\tau^{(l)} = \frac{F_\tau^{(l)}}{m}, \quad \hat{\mathbf{v}}_t^{(l)} = \mathbf{v}_0^{(l)} + \int_0^t \mathbf{a}_\tau^{(l)} d\tau, \quad \hat{\mathbf{x}}_t^{(l)} = \mathbf{x}_0^{(l)} + \int_0^t \hat{\mathbf{v}}_\tau^{(l)} d\tau. \quad (2)$$

Case 1: non-simultaneous integration. In this case, we need to call the ODE/SDE function twice. The first ODE/SDE function call corresponds to solving the acceleration integration with $[d\mathbf{v}/dt] = [F/m]$. This means that if we are integrating for T timesteps, then we have

$$\begin{aligned} \mathbf{v}_1 &= \text{ODE/SDE Integration}(\text{model}(\mathbf{x}_0)/m), \\ \mathbf{v}_2 &= \text{ODE/SDE Integration}(\text{model}(\mathbf{x}_1)/m), \\ &\dots \\ \mathbf{v}_T &= \text{ODE/SDE Integration}(\text{model}(\mathbf{x}_{T-1})/m). \end{aligned} \quad (3)$$

The second ODE/SDE call function call corresponding to solving the velocity integration with $[d\mathbf{x}/dt] = [\mathbf{v}]$. Similarly, this means that if we are integrating for T timesteps, then we have:

$$\begin{aligned} \mathbf{x}_1 &= \text{ODE/SDE Integration}(\mathbf{v}_0), \\ \mathbf{x}_2 &= \text{ODE/SDE Integration}(\mathbf{v}_1), \\ &\dots \\ \mathbf{x}_T &= \text{ODE/SDE Integration}(\mathbf{v}_{T-1}). \end{aligned} \quad (4)$$

Thus, we will call the ODE/SDE integration calls **twice**, as in Equations (3) and (4).

Case 2: simultaneous integration. The case means that we can call the acceleration integration and velocity integration simultaneously, as shown in Equation 14 of the manuscript, we utilize

$$\begin{bmatrix} d\mathbf{x}/dt \\ d\mathbf{v}/dt \end{bmatrix} = \begin{bmatrix} \mathbf{v} \\ F/m \end{bmatrix}. \quad (5)$$

Then we call the integration for T timesteps as:

$$\begin{aligned} \begin{bmatrix} \mathbf{v}_1 \\ \mathbf{x}_1 \end{bmatrix} &= \text{ODE/SDE Integration} \left(\begin{bmatrix} \text{model}(\mathbf{x}_0)/m \\ \mathbf{v}_0 \end{bmatrix} \right), \\ \begin{bmatrix} \mathbf{v}_2 \\ \mathbf{x}_2 \end{bmatrix} &= \text{ODE/SDE Integration} \left(\begin{bmatrix} \text{model}(\mathbf{x}_1)/m \\ \mathbf{v}_1 \end{bmatrix} \right), \\ &\dots \\ \begin{bmatrix} \mathbf{v}_T \\ \mathbf{x}_T \end{bmatrix} &= \text{ODE/SDE Integration} \left(\begin{bmatrix} \text{model}(\mathbf{x}_{T-1})/m \\ \mathbf{v}_{T-1} \end{bmatrix} \right). \end{aligned} \quad (6)$$

Thus, we will call the ODE/SDE integration calls just **once**, in comparison to twice in the first case (Equations (3) and (4)), resulting in a twofold increase in efficiency.

We added this in Section E.3 in the revised Supplementary Information.

Reviewer Comment 1.6 — 6. About Efficiency Why does the NeuralMD algorithm reduce the need for discretization compared to the DenoisingLD method, enabling conformational prediction with fewer time steps? Please provide a detailed explanation.

Reply: Thank you for raising the question on efficiency. We briefly described this in the previous manuscript, and we revised it in Section 2.7 in the latest manuscript. Here is a polished revision:

Since all ML methods share a similar backbone architecture (BindingNet and its variants), their efficiency differences arise from their computational approaches and hyperparameters. GNN-MD is the most efficient, as it requires no integration steps. NeuralMD incorporates an augmented integration step, with the optimal step size set to 0.5 of the snapshot interval, *i.e.*, two integration steps are performed to predict the next snapshot, making it slightly slower than VerletMD. In contrast, DenoisingLD is the slowest method, as it uses a smaller optimal timestep of one-tenth the snapshot interval, requiring 10 integration steps to generate the next snapshot during inference.

Reviewer Comment 1.7 — 7. Pretraining. Can pretraining on data annotated with energy/force labels further improve the performance?

Reply: Thank you for raising this question on pretraining. Firstly, a brief answer to your question is that: yes, pretraining on energy/force prediction should be helpful. We have an ongoing project that is doing this.

Meanwhile, we would like to mention the key differences in energy/force prediction and trajectory prediction.

- In our current problem formulation, we assume that only positional data is available. This follows the **ML for Trajectory Learning for MD Simulation** research line of several DenoisingLD papers (CGDMS [11], DiffMD [12], DFF [13], CG-MD [14]), as discussed in Section A.2.
- Therefore, incorporating additional supervised labels such as energy or force would define a different problem setting, as discussed in **ML for Potential Energy and Force Learning for MD Simulation** in Section A.2. That said, if the objective is to address the general problem of dynamics simulation, incorporating such additional data—like energy or force labels—would indeed be beneficial.

Reviewer Comment 1.8 — 8. Suggestions for Improving Figures and Names. The information provided in Figure 1 is limited, as it only presents the formula for the Velocity Verlet algorithm. It is recommended to enhance the figure by including additional information, such as comparisons with GNN-MD and DenoisingLD. This would help emphasize the innovative aspects of the proposed method. The design and color scheme of Figure 2a could be improved for better aesthetics and clarity.

Reply: Thank you for raising this comment. We have revised the figure, and the revision is in Figure 2 below. We also uploaded the revised version in the manuscript. We would be happy to further refine the figures based on any additional suggestions you may have.

Figure 2: Illustrations of binding dynamics. The landscape depicts the energy level, and the binding dynamic leads to an equilibrium state with lower energy. (a) Pipeline for second-order methods, Velocity Verlet and NeuralMD (ours). (b) Pipeline for GNN-MD. (c) Pipeline for Denoising-LD.

Reviewer Comment 1.9 — Besides, I suggest the authors consider using another name of BindingNet. Because it has been used by a protein-ligand complex data, which may easily cause confusions.(Remarks on code availability)

Reply: Thank you for your suggestion. We will rename BindingNet to FrameNet in the final version.

Reviewer 2

We would like to express our gratitude to Reviewer 2 for raising important questions regarding dataset clarification and problem formulation. These are indeed limitations of the current NeuralMD work, as discussed in Section 3 of the manuscript. We consider them valuable directions for future exploration and provide further details in our responses below.

Reviewer Comment 2.1 — 1. It is not clear that with the NeuralMD, one can perform interaction analysis throughout trajectories.

Reply: Thank you for raising this comment on the interaction analysis.

First, we would like to mention that NeuralMD has certain limitations, as we discussed in Section 3 of the manuscript, including the dataset and GPU limitations for the flexible setting.

However, NeuralMD still possesses the ability for certain primary interaction analysis:

- We provide a case study on 4G3E, as described in the manuscript Section 2.8, as well as in Comment 1.4.
- We provide the demonstration for 10 binding complex examples in the supplementary files (also on the website).

Reviewer Comment 2.2 — 2. As the authors point out, proteins being fixed during the simulation is a major drawback, as one usually resorts to MD simulations for exactly the cases in which protein structures are flexible. This assumption severely limits the utility of the method in drug discovery.

Reply: Thank you for raising this important question. In our current setup, we assume the protein remains fixed for the following reasons:

- This is due to the large volume of proteins.
- Modeling both the ligand and protein and applying the second-order integration for simulating the dynamics goes above the GPU memory limitation.
- This assumption is also common in the top-tier community, *e.g.*, DiffDock [1]. BTW. We also found that the DiffDock-PP [2] shared by Reviewer 1 using the inflexible setting, and they claimed that “This assumption is often realistic (Vakser, 2014) and even leads to improved results for most interacting proteins (Desta et al., 2020).”

We would like to highlight that memory efficiency is currently the primary limitation, and addressing it is a key focus of our future work. In particular, our group is actively exploring geometric modeling techniques that better capture molecular rigidity, which can also be naturally extended to model protein residues more effectively.

Reviewer Comment 2.3 — 3. Training dataset is in the order of 10,000, which doesn't seem to be enough. I wonder if one can augment it by generating protein-ligand complex structures by docking.

Reply: Thank you for pointing out this issue. We acknowledge that the dataset insufficiency is one inherent limitation. To the best of our knowledge, MISATO [3] is by far the largest one with semi-quantum methods. Of course, we could try the docking algorithm, yet it will be another research direction on dataset generation, along the MISATO. We mentioned this as one limitation for future exploration in Section 3 of the manuscript.

Reviewer 3

We would like to express our gratitude to Reviewer 3 for raising questions on the problem formulation and RMSF evaluation. We have carefully addressed each of these points in detail, as explained below.

Reviewer Comment 3.1 — What is the main purpose of this work? Do they want to predict the binding processes or just the binding poses? Since they often use the word, “trajectories”, like MD simulations, I was confused for their main research purposes.

Reply: Thank you for raising this question. Our goal is to predict the trajectories of the protein-ligand binding processes. We have mentioned this in Section 2.1 **Problem Setting** and Section 4 **Methods** of the manuscript.

Reviewer Comment 3.2 — 2) Initial structures. According to the movies on their github, the motions of ligand are relatively small. Though they are flexible, their positions relative to the protein do not change so much. What are the initial structures in the predictions? If they separated a protein and a ligand, how they can predict the protein-ligand complexes? What are their target conformational spaces to be sampled by ML?

Reply: The initial positions are provided in the MISATO dataset.

In the MISATO dataset [3], they simulate the trajectories for 10ns, and drop the first 2ns. We take the position at the first timestep as the input, and the goal is to predict the remaining positions along the MD trajectories.

Reviewer Comment 3.3 — 3) Binder and non-binder distinctions. Can they distinct binder ligands or non-binder ligands in their methods?

Reply: Thank you for raising this question. The problem definition of our NeuralMD paper is $p(\text{ligand positions at } t+1, \dots \mid \text{initial positions at } t)$.

The binder and non-binder distinction is another task: $F(\text{ligand, protein}) = \text{True/False}$.

Reviewer Comment 3.4 — 4) RMSF in Figure 3 What is the ground-truth RMSF? Is it MD simulation result or B-factor in X-ray structure? Why the RMSF in ML is generally so small? What is the physical meaning of the RMSF? Why the ground-truth RMSF in 1KTI is so large?

Reply: The ground-truth RMSF is provided by the MISATO dataset [3], which is the MD simulation result.

RMSF meaning. RMSF is short for Root Mean Square Fluctuation. It quantifies the mean square deviation of each atom’s position from its average position over the trajectory, providing a measure of

positional fluctuations. Or formally, it is defined as:

$$\sqrt{\mathbb{E}_t \left[\frac{1}{N} \sum_{i=1}^N \|r_{i,t} - \langle r_i \rangle\|^2 \right]}, \quad (7)$$

where $r_{i,t}$ is the i -th atom position at time t (ground truth or sampled by ML methods) and $\langle r_i \rangle$ is the average position for the i -th atom across time. We have this detailed in Section 2.6 in the manuscript.

Why RMSF in ML methods is small. That is the generalization ability of ML models. In Figure 3, the experiment setting corresponds to the **Generalization of One Single Trajectory**. In other words, for each trajectory, we train NeuralMD on the first 80 snapshots (positions) and generalize to the remaining snapshots (the remaining positions along the MD trajectory). The RMSF values are small and tend to be smaller because the first 80 snapshots fluctuations are smaller.

RMSF in 1KTI. We have created demonstrations for the 10 binding complex examples in the supplementary files (also on the website). Notably, there is a significant jump for 1KT1 (ground-truth), which is due to the simulation results from the MISATO dataset [3].

References

- [1] G. Corso, H. Stärk, B. Jing, R. Barzilay, and T. Jaakkola, “Diffdock: Diffusion steps, twists, and turns for molecular docking,” *arXiv preprint arXiv:2210.01776*, 2022.
- [2] M. A. Ketata, C. Laue, R. Mammadov, H. Stärk, M. Wu, G. Corso, C. Marquet, R. Barzilay, and T. S. Jaakkola, “Diffdock-pp: Rigid protein-protein docking with diffusion models,” *arXiv preprint arXiv:2304.03889*, 2023.
- [3] T. Siebenmorgen, F. Menezes, S. Benassou, E. Merdivan, K. Didi, A. S. D. Mourão, R. Kitel, P. Liò, S. Kesselheim, M. Piraud *et al.*, “Misato: machine learning dataset of protein–ligand complexes for structure-based drug discovery,” *Nature Computational Science*, pp. 1–12, 2024.
- [4] V. G. Satorras, E. Hoogeboom, and M. Welling, “E (n) equivariant graph neural networks,” in *International conference on machine learning*. PMLR, 2021, pp. 9323–9332.
- [5] J. Jumper, R. Evans, A. Pritzel, T. Green, M. Figurnov, O. Ronneberger, K. Tunyasuvunakool, R. Bates, A. Žídek, A. Potapenko *et al.*, “Highly accurate protein structure prediction with alphafold,” *nature*, vol. 596, no. 7873, pp. 583–589, 2021.
- [6] N. Thomas, T. Smidt, S. Kearnes, L. Yang, L. Li, K. Kohlhoff, and P. Riley, “Tensor field networks: Rotation-and translation-equivariant neural networks for 3d point clouds,” *arXiv preprint arXiv:1802.08219*, 2018.
- [7] J. Yim, B. L. Trippe, V. De Bortoli, E. Mathieu, A. Doucet, R. Barzilay, and T. Jaakkola, “Se (3) diffusion model with application to protein backbone generation,” *arXiv preprint arXiv:2302.02277*, 2023.
- [8] J. Yim, A. Campbell, A. Y. Foong, M. Gastegger, J. Jiménez-Luna, S. Lewis, V. G. Satorras, B. S. Veeling, R. Barzilay, T. Jaakkola *et al.*, “Fast protein backbone generation with se (3) flow matching,” *arXiv preprint arXiv:2310.05297*, 2023.
- [9] A. J. Bose, T. Akhound-Sadegh, G. Huguet, K. Fatras, J. Rector-Brooks, C.-H. Liu, A. C. Nica, M. Korablyov, M. Bronstein, and A. Tong, “Se (3)-stochastic flow matching for protein backbone generation,” *arXiv preprint arXiv:2310.02391*, 2023.
- [10] G. Huguet, J. Vuckovic, K. Fatras, E. Thibodeau-Laufer, P. Lemos, R. Islam, C.-H. Liu, J. Rector-Brooks, T. Akhound-Sadegh, M. Bronstein *et al.*, “Sequence-augmented se (3)-flow matching for conditional protein backbone generation,” *arXiv preprint arXiv:2405.20313*, 2024.
- [11] J. G. Greener and D. T. Jones, “Differentiable molecular simulation can learn all the parameters in a coarse-grained force field for proteins,” *PloS one*, vol. 16, no. 9, p. e0256990, 2021.
- [12] F. Wu and S. Z. Li, “Diffmd: A geometric diffusion model for molecular dynamics simulations,” in *Proceedings of the AAAI Conference on Artificial Intelligence*, vol. 37, no. 4, 2023, pp. 5321–5329.

- [13] M. Arts, V. Garcia Satorras, C.-W. Huang, D. Zugner, M. Federici, C. Clementi, F. Noe, R. Pinsler, and R. van den Berg, “Two for one: Diffusion models and force fields for coarse-grained molecular dynamics,” *Journal of Chemical Theory and Computation*, 2023.
- [14] X. Fu, T. Xie, N. J. Rebello, B. D. Olsen, and T. Jaakkola, “Simulate time-integrated coarse-grained molecular dynamics with geometric machine learning,” *arXiv preprint arXiv:2204.10348*, 2022.
- [15] K. M. Pflug and R. Sitcheran, “Targeting nf- κ b-inducing kinase (nik) in immunity, inflammation, and cancer,” *International journal of molecular sciences*, vol. 21, no. 22, p. 8470, 2020.
- [16] H. Teng, L. Xue, Y. Wang, X. Ding, and J. Li, “Nuclear factor κ b-inducing kinase is a diagnostic marker of gastric cancer,” *Medicine*, vol. 99, no. 5, p. e18864, 2020.
- [17] G. de Leon-Boenig, K. K. Bowman, J. A. Feng, T. Crawford, C. Everett, Y. Franke, A. Oh, M. Stanley, S. T. Staben, M. A. Starovasnik *et al.*, “The crystal structure of the catalytic domain of the nf- κ b inducing kinase reveals a narrow but flexible active site,” *Structure*, vol. 20, no. 10, pp. 1704–1714, 2012.

Revisions made to manuscript NCOMMS-24-74647

We would like to extend our sincere appreciation to the editor and reviewers for their invaluable comments and suggestions, which have significantly enhanced the quality of this manuscript.

In response, we want to highlight **four** key points:

1. We conducted ablation studies by changing the three granularities to two granularities.
2. We modified the statements on rigidity and acceleration of using NeuralMD.
3. We clarified the main usage of NeuralMD, which is complementary to the existing classic MD methods.

To summarize, while we acknowledge the current limitations of NeuralMD, we believe it addresses important challenges in the community and offers a valuable step forward. We are grateful for the constructive feedback, and we have meticulously revised the manuscript (marked in green) in accordance with the insightful comments. We would also be grateful if they could recognize the contributions of our proposed multi-grained vector frame modeling approach. Please find our detailed responses below.

Reviewer 1

Thank you for acknowledging our revisions to the statements, figures, and methods – particularly our clarifications regarding the distinction between energy/force and MD tasks, the explanation of pretraining, and the subsequent experimental validation plan. Your remaining question on SE(3)-equivariance has been addressed below.

Reviewer Comment 1.1 — I also thank the author for the explanations regarding SE(3) equivariance, E(3) equivariance, and reflection-antisymmetry in the network. However, even if these methods don't explicitly emphasize the concepts of SE(3) equivariance and reflection-antisymmetry, they also focus on multi-scale modeling of protein-ligand interactions, which is a central aspect of this work. Without a comparison, it's difficult to assess the practical significance of the proposed network's architectural innovations.

Therefore, I encourage the authors to conduct such comparisons and provide corresponding discussions or empirical analyses to better highlight the novelty and effectiveness of their design.

Reply:

Thank you for raising this question on multi-scale modeling. In comparison, we list the main differences of the baselines below. One thing we would like to highlight is that most of these baselines (e.g., FABind, MolCRAFT) are parallel work and their underlying physics is wrong. They applied a simple paradigm as the interaction modeling, which we added as an ablation study.

- EquiBind is just applying E(3)-equivariant on ligands and proteins for the binding prediction. There is no multi-scaling modeling. link.

(a) Three granularities of vector frame basis in NeuralMD, Ours

(b) Two granularities of vector frame basis in NeuralMD, Ablation

Figure 1: Fig. (a) corresponds to the three granularities of vector frame basis in NeuralMD (Ours). Fig. (b) lists the three granularities of vector frame basis in the ablation studies, *i.e.*, NeuralMD (Ablation). The main difference lies in the third granularity (if we take (b.3) as one): Fig. (a.3) utilizes the backbone atoms in pockets to model the binding complex, while Fig. (b.3) considers all the backbone atoms and ligand atoms for frame construction.

- FABind is applying $E(3)$ -equivariant on two hierarchies, link. Besides, this is a parallel work.
 - It includes a pocket prediction module to determine the binding site. Note that this is not required in our case, since we include the ligand information as the initial input.
 - Then goes to the FABind layer, which is similar to the EquiBind layer, using $E(3)$ -equivariant layers on ligands and proteins for their interaction prediction.
- MolCRAFT, directly models the ligand atoms and protein atoms, *i.e.*, no hierarchical or multi-scale modeling, link. Besides, this is also a parallel work.

Both EquiBind and FABind perform message passing between ligand and receptor, using either joint-then-separate or separate-then-joint multi-scale modeling. In contrast, our NeuralMD introduces a three-level hierarchical design. As shown in Figure 1, Fig. (a) illustrates the three vector frame granularities in NeuralMD (Ours), while Fig. (b) shows the ablation version. The key difference is in the third granularity (if we take (b.3) as one): Fig. (a.3) uses pocket backbone atoms, whereas Fig. (b.3) includes all backbone and ligand atoms for frame construction.

We conducted an ablation study of the new NeuralMD architecture using 10 single-traj experiments, and the results are listed in Table 1. We mark our proposed hierarchical structure (Fig. a in Figure 1) as **NeuralMD (ODE), Ours** and **NeuralMD (SDE), Ours**. We mark the structures in ablation studies (Fig. b in Figure 1) as **NeuralMD (SDE), Ablation** and **NeuralMD (SDE), Ablation**. For the general performance, the hierarchical structures obtain (ours) obtain better performance than the non-hierarchical architectures (ablation studies).

Table 1: Results on ten single-trajectory binding dynamics predictions in the semi-flexible setting. Results with optimal training loss are reported. Four evaluation metrics are considered: MAE (\AA , \downarrow), MSE (\downarrow), Matching(\downarrow), and Stability ($\%$, \uparrow).

		NeuralMD (ODE), Ours	NeuralMD (ODE), Ablation	NeuralMD (SDE), Ours	NeuralMD (SDE), Ablation
5WIJ	MAE (\downarrow)	2.252	2.259	2.260	2.600
	MSE (\downarrow)	1.514	1.506	1.514	1.766
	Matching (\downarrow)	0.464	0.509	0.615	0.735
	Stability (\uparrow)	82.046	77.125	67.464	57.138
4ZX0	MAE (\downarrow)	1.878	1.879	2.158	2.156
	MSE (\downarrow)	1.263	1.255	1.455	1.455
	Matching (\downarrow)	0.428	0.475	0.696	0.886
	Stability (\uparrow)	81.401	77.489	59.109	52.255
3EOV	MAE (\downarrow)	3.858	3.624	3.395	3.435
	MSE (\downarrow)	2.651	2.471	2.309	2.350
	Matching (\downarrow)	1.062	1.160	0.962	1.095
	Stability (\uparrow)	47.328	42.652	50.108	44.788
4K6W	MAE (\downarrow)	3.656	7.933	3.765	4.049
	MSE (\downarrow)	2.400	5.204	2.501	2.650
	Matching (\downarrow)	0.928	10.751	1.076	1.352
	Stability (\uparrow)	49.438	41.319	49.700	42.515
1KTI	MAE (\downarrow)	6.675	6.706	6.646	6.664
	MSE (\downarrow)	4.176	4.181	4.141	4.163
	Matching (\downarrow)	0.337	0.303	0.167	0.214
	Stability (\uparrow)	86.430	87.932	98.508	95.103
1XP6	MAE (\downarrow)	1.924	1.929	2.061	2.119
	MSE (\downarrow)	1.280	1.285	1.356	1.403
	Matching (\downarrow)	0.537	0.588	0.615	0.768
	Stability (\uparrow)	75.533	70.409	69.423	62.079
4YUR	MAE (\downarrow)	6.957	6.975	7.038	6.977
	MSE (\downarrow)	4.597	4.607	4.679	4.593
	Matching (\downarrow)	0.584	0.700	0.749	0.820
	Stability (\uparrow)	69.775	64.365	60.344	55.515
4G3E	MAE (\downarrow)	2.191	2.672	2.345	2.365
	MSE (\downarrow)	1.453	1.771	1.536	1.564
	Matching (\downarrow)	0.505	0.698	0.521	0.594
	Stability (\uparrow)	71.436	65.011	68.729	62.840
6B7F	MAE (\downarrow)	3.921	3.952	3.842	3.974
	MSE (\downarrow)	2.652	2.660	2.601	2.691
	Matching (\downarrow)	0.459	0.543	0.741	0.622
	Stability (\uparrow)	75.692	71.733	57.917	63.308
3B9S	MAE (\downarrow)	3.039	2.990	3.132	3.390
	MSE (\downarrow)	1.999	1.995	2.078	2.263
	Matching (\downarrow)	0.659	1.055	0.444	0.453
	Stability (\uparrow)	76.065	56.667	77.801	75.093

Reviewer 2

We sincerely thank Reviewer 2 for the thoughtful evaluation and encouraging comments. We appreciate your recognition of NeuralMD’s potential utility in current drug discovery practices, as well as your acknowledgment of our detailed response. We also agree that there remains room for improvement, and we see this as an exciting direction for future research. Thank you again for recommending our manuscript for publication.

Reviewer 3

We would like to express our gratitude to Reviewer 3 for raising concerns on the clarification of the limitation, the RMSF annotation, as well as the detailed discussion on comparison with the classic MD methods. We have revised the statements and clarified them in the revised manuscript. Please check the details below.

Reviewer Comment 3.1 —

The author needs to emphasize the limitation of the current model in the abstract and maintext and discuss the future perspectives about the flexible treatment of protein structures.

Reply: Thank you for highlighting the potential impact of the semi-flexible/rigidity assumption. We have removed the unclear statement and highlighted the rigidity assumption in the revised manuscript and marked in green.

Reviewer Comment 3.2 — Also, the physical meaning of RMSF shown in Figure 3 is different from the RMSF in MD simulations, because of the different treatment of protein flexibility. If such quantities are shown without reasonable explanations, the data will confuse readers of this manuscript.

Reply: Thank you for raising this question. In our project, we are calculating the RMSF only considering the fluctuations of the ligands, and we rename it to **RMSF-Ligand** in the revised manuscript and marked in green.

Reviewer Comment 3.3 — Also, the treatment of solvent is different between the ML surrogate model and the MD simulation. In the MD simulation, solvent molecules are explicitly included, while the ML surrogate does not.

I suggest that the author compared these two different methods with the same conditions and approximations. Otherwise, just the numerical comparison of the computational time is not meaningful.

Reply:

Thank you for bringing this up. It's an important point, and we appreciate the opportunity to clarify.

As mentioned, the training dataset for NeuralMD is based on the MISATO dataset [1]. The molecular dynamics (MD) simulations in MISATO were carried out using Amber20, where the protein–ligand complexes were solvated in the TIP3P water model and neutralized with Na⁺ and Cl⁻ ions [1]. As a result, NeuralMD was trained to model protein–ligand dynamics under aqueous conditions. When applied to other complexes in the test set, its predictions implicitly assume similar solvent environments, i.e., in an in-distribution setting (in contrast to out-of-distribution). This is important when interpreting the results.

In other words, NeuralMD and Amber20/GROMACS are different in the underlying principles.

- NeuralMD is a machine learning (ML) model trained on MD trajectories of protein–ligand complexes in explicit solvent, which are typically generated from traditional MD methods like Amber/GROMACS. Its simulations/predictions reflect patterns learned from such data, with solvent effects implicitly captured from the training dataset rather than explicitly simulated.
- Additionally, the NeuralMD SDE (Eq. 11 in the manuscript) includes a stochastic term that implicitly captures the effects of the solvent environment, even though explicit solvent information is not available in the MISATO dataset. If such solvent information were provided, it could be directly incorporated into the modeling. However, this remains a limitation of the current dataset.

- In contrast, Amber20 and GROMACS are traditional physics-based MD engines that use defined force fields and explicitly model solvent molecules under user-specified simulation conditions.

Given these differences, we believe it may not be appropriate to directly apply traditional MD parameter settings to NeuralMD. Nevertheless, we fully agree with your suggestion: NeuralMD would benefit from additional training data generated under well-controlled MD protocols to further improve its simulation/prediction accuracy.

Regarding the comparison of different methods under the same conditions and approximations, we ensured that the settings were aligned. While the training data of NeuralMD come from Amber20-based MD simulations (with TIP3P water, as well as Na⁺ and Cl⁻ ions neutralization), we used GROMACS for benchmark simulations under comparable conditions for the protein-ligand complex, applying the AMBER99SB-ILDN force field, solvating with TIP3P water, and neutralizing with ions.

References

- [1] T. Siebenmorgen, F. Menezes, S. Benassou, E. Merdivan, K. Didi, A. S. D. Mourão, R. Kitel, P. Liò, S. Kesselheim, M. Piraud *et al.*, “Misato: machine learning dataset of protein–ligand complexes for structure-based drug discovery,” *Nature Computational Science*, pp. 1–12, 2024.